# Improving Branching in Neural Network Verification with Bound Implication Graph

## Abstract

Many state-of-the-art neural network verifiers for ReLU networks rely on branch-and-bound (BaB)-based methods. They branch ReLUs into positive (active) and negative (inactive) parts, and bound each subproblem independently. Since the cost of verification heavily depends on the number of subproblems, reducing the total number of branches is the key to verifying neural networks efficiently. Implications among neurons can eliminate subproblems: for example, when one or more ReLU neurons are branched into the active (or inactive) case, they may imply that a set of other neurons from any layers become active or inactive. In this paper, we propose a novel optimization formulation to find these implications, and a scalable method to solve these optimization problems among all neurons within tens of seconds, even for large ResNets, by reusing pre-computed variables in popular bound-propagation-based verification methods such as $\alpha$-CROWN. Our method is less restrictive than previous approaches and can produce significantly more bound implications compared to prior work, which may benefit many BaB-based verifiers. When evaluated on a set of popular verification benchmarks and a new benchmark consisting of harder verification problems, we consistently reduce the verification time and verify more problems than state-of-the-art verification tools.

## 1 Introduction

As deep neural networks (DNNs) have been widely adopted in safety-critical domains (Kiran et al., 2021; Popova et al., 2018; Cao et al., 2021; Eykholt et al., 2018), providing rigorous verification for their robustness, safety and correctness is critical. To address these concerns, neural network verification algorithms (Ehlers, 2017; Katz et al., 2017; Tjeng et al., 2019) have been developed to prove certain properties of a DNN rigorously. A traditional example is certifying robustness under adversarial perturbations, where one aims to verify that the classification outcome does not change under bounded input perturbations. More general specifications of DNNs can also be proved, and DNN verification has become a valuable tool in different domains such as control (Everett, 2021) and cyberphysical systems (Rober et al., 2022; Harapanahalli et al., 2023). Proving these specifications typically requires rigorously bounding the outputs of DNNs considering input perturbations.

One popular DNN verification approach is the Branch and Bound (BaB) based methods (Bunel et al., 2018; 2020; Palma et al., 2021; Bak et al., 2020; Wang et al., 2018a;b; 2021), which exploits the piece-wise-linear property of ReLU-based DNNs. Since $\text{ReLU}(x) = \max\{x, 0\}$ consists of two linear pieces, the verification problem can be split into two easier subproblems, each with one less nonlinear ReLU neuron. This property is the foundation for BaB-based verification approaches and many state-of-the-art verifiers, such as $\alpha,\beta$-CROWN (Wang et al., 2021; Xu et al., 2020b; Zhang et al., 2022), MN-BaB (Ferrari et al., 2021), and VeriNet (Henriksen & Lomuscio, 2020), utilized BaB as their core verification procedure.

Specifically, a BaB-based verifier splits a ReLU neuron into inactive (inputs $\leq 0$) and active (input $> 0$) cases (**Branching**), creating two subproblems after each neuron is split; then the lower and upper bounds of the model output for each subproblem are calculated (**Bounding**). The bounding process for each subproblem typically involves solving a relaxed optimization problem, which is often the most costly step. After one step of BaB, the output bounds usually improve and may lead to successful verification of subproblems. If the bounds are still not tight enough, the verifier will recursively choose the next ReLU neuron to split in each unsolved subproblem.

The key to improving the efficiency of BaB-based verifiers is to reduce the number of subproblems created. Several works (Kouvaros & Lomuscio, 2021; Botoeva et al., 2020) considered the correlations (implications) among ReLU neurons: when a neuron is branched into the active (or inactive) case, it may imply the active (or inactive) status of other neurons; a graph can then be formed based on these bound implications (dependencies). Although bound implications may exist between any two neurons (e.g., a ReLU in the last layer of NN is active may imply that a ReLU in the first layer should always be inactive), existing work such as (Kouvaros & Lomuscio, 2021; Botoeva et al., 2020) can only detect some limited form of implications between neurons, such as among neurons in the same layer or in consecutive layers, and simple approaches such as interval bounds are used to find implications.

In our work, we first formulate the problem of finding bound implications as an optimization problem, which leads to a stronger formulation than previous works, and opportunities to find many more implications. This problem involves a challenging non-convex constraint, and we must avoid the usage of linear programming (LP) or mixed integer programming (MIP) solver because of the large number of pairs of ReLU neurons involved in finding these implications. To address this challenge, we novelly reuse the pre-computed variables in existing bound-propagation-based verification methods such as $\alpha$-CROWN and solve a simple LP problem either in closed form solution or via gradient descent, *without* relying on an expensive LP solver. We can quickly find hundreds to thousands of bound implications between neurons using our proposed approach, significantly more compared to previous approaches. In addition, by utilizing common split constraints in BaB, we can build multiple bound implication graphs (BIGs), each for a subset of subproblems, to further improve the effectiveness of bound implications. Our main technical contributions are:

- We formulate the process of finding bound implications as an optimization problem, which allows us to find correlations between two neurons from arbitrary layers, not restricted to implications in the same or consecutive layers in previous work;

- We propose a novel approach to reuse information from already computed in bound propagation (e.g., $\alpha$-CROWN) to quickly solve these optimization problems to find bound implications between thousands of pairs of ReLU neurons within a few seconds. Combined with a unique filtering strategy, our approach is scalable to large DNNs (such as 19 layer ResNets on TinyImageNet).

- Our optimization formulation is much stronger compared to existing simple approaches to find bound implications: we found 7 to 170 times more implications among neurons on average than previous work (Kouvaros & Lomuscio, 2021; Botoeva et al., 2020).

- When evaluated on robustness verification benchmarks on MNIST, CIFAR-10, CIFAR-100, and TinyImageNet datasets, including new benchmarks consisting of hard verification problems, we save up to 37% subproblems during BaB, consistently reduce the verification time, and verify more problems compared to baseline BaB methods such as $\beta$-CROWN and MN-BaB.

## 2 BACKGROUND

DNN verification methods are developed to rigorously prove formal specifications on DNNs. We let $\boldsymbol{x} \in \mathcal{X} \subseteq \mathbb{R}^{d_0}$ to denote the input, and let $f : \mathbb{R}^{d_0} \to \mathbb{R}$ to denote a neural network. For an $L$-layer neural network, $f$ is defined by $f(\boldsymbol{x}) = \boldsymbol{z}^{(L)}(\boldsymbol{x})$, where $\boldsymbol{z}^{(i)}(\boldsymbol{x}) = \boldsymbol{W}^{(i)}\hat{\boldsymbol{z}}^{(i-1)}(\boldsymbol{x}) + \boldsymbol{b}^{(i)}$ and $\hat{\boldsymbol{z}}^{(i)}(\boldsymbol{x}) = \sigma(\boldsymbol{z}^{(i)}(\boldsymbol{x}))$, where $i \in [L] := \{1, \cdots, L\}$. Specifically, $\hat{\boldsymbol{z}}^{(0)}(\boldsymbol{x}) = \boldsymbol{x}$. $\sigma$ is the activation function and we focus on ReLU in this paper. We call $z_j^{(i)}(\boldsymbol{x})$ (in short as $z_j^{(i)}$) pre-activation and $\hat{z}_j^{(i)}(\boldsymbol{x})$ (in short as $\hat{z}_j^{(i)}$) post-activation values of the $j$-th neuron in the $i$-th layer, $j \in [d_i]$ and $d_i$ is the number of neurons of the $i$-th layer. Canonically, verification specifications can be written as proving $f(\boldsymbol{x}) > 0, \forall \boldsymbol{x} \in \mathcal{C}$; i.e., we say $f(\boldsymbol{x})$ is verified if $f(\boldsymbol{x}) > 0, \forall \boldsymbol{x} \in \mathcal{C}$. DNN verification methods solve the following optimization problem:

$$\min f(\boldsymbol{x}) = \boldsymbol{z}^{(L)}(\boldsymbol{x}) \quad \text{s.t.} \ \boldsymbol{z}^{(i)} = \boldsymbol{W}^{(i)}\hat{\boldsymbol{z}}^{(i-1)} + \boldsymbol{b}^{(i)}, \hat{z}^{(i)} = \sigma(\boldsymbol{z}^{(i)}), \boldsymbol{x} \in \mathcal{C}, i \in [L-1]. \quad (1)$$

The $\mathcal{C}$ defines the input region, and we focus on $\ell_\infty$-bounded perturbations following the literature (Wong & Kolter, 2018; Xu et al., 2020b; Wang et al., 2021; Müller et al., 2022b). Therefore, given a clean input instance $\boldsymbol{x}_0$ and the perturbation magnitude $\epsilon$, $\mathcal{C} := \{\boldsymbol{x} \mid \|\boldsymbol{x} - \boldsymbol{x}_0\|_\infty \le \epsilon\}$.

**Bound propagation methods.** Bound propagation methods like CROWN (Zhang et al., 2018) and $\beta$-CROWN (Wang et al., 2021) aim to calculate a lower bound for $f_L(\boldsymbol{x})$ without using the LP solver:

$$\min_{\boldsymbol{x} \in \mathcal{C}} f_L(\boldsymbol{x}) \ge \min_{\boldsymbol{x} \in \mathcal{C}} \mathbf{a}^{(i)^\top} \hat{\boldsymbol{x}}^{(i)} + c^{(i)} \quad (2)$$

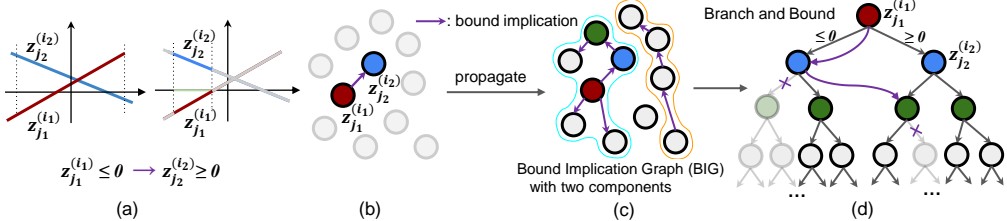

Figure 1: (a) Bound implications: based on the linear lower bounds of two neurons, if one neuron is set to inactive (<0), the other neuron will always be active (>0) under constraints; (b) the two neurons are connected by an edge on the Bound Implication Graph (BIG), and (c) more bound implications create a larger BIG with multiple components. (d) During branch and bound (BaB), BIG is used to remove unnecessary subproblems, so fewer subproblems are needed and verification time is saved.

With $i = L - 1$ the above holds trivially with $\mathbf{a}^{(L-1)} = \mathbf{W}^{(L)}$, $c^{(L-1)} = \mathbf{b}^{(L)}$. Propagating bounds from the $L$-layer to the previous layer repeatedly will eventually reach the input layer:

$$\min_{\boldsymbol{x} \in \mathcal{C}} f_L(\boldsymbol{x}) \geq \min_{\boldsymbol{x} \in \mathcal{C}} {\mathbf{a}^{(0)}}^\top \boldsymbol{x} + c^{(0)} \tag{3}$$

This linear optimization problem can be easily solved to provide a lower bound for $f(\boldsymbol{x})$. Eq. (3) are usually also applied to intermediate layer neurons. In our paper, we will reuse the variables $\mathbf{a}$, $c$ which are already available during bound propagation to build our bound implication graph efficiently.

**Active, inactive, and unstable neurons.** To leverage the branch-and-bound technique, we need to know the pre-activation bounds $\boldsymbol{l}^{(i)}$ and $\boldsymbol{u}^{(i)}$ for the inputs of each ReLU layer, such that $\boldsymbol{l}^{(i)} \leq \boldsymbol{x}^{(i)} \leq \boldsymbol{u}^{(i)}$. Those bounds are valid for $\boldsymbol{x} \in \mathcal{C}$, and can be obtained via cheap bound propagation methods such as IBP (Gowal et al., 2019), CROWN (Zhang et al., 2018) or $\alpha$-CROWN (Xu et al., 2020b). Then ReLU neurons for each layer $i$ can be classified into three classes (Wong & Kolter, 2018): $l_j^{(i)} \geq 0$ (active neurons), $u_j^{(i)} \leq 0$ (inactive neurons), and $l_j^{(i)} \leq 0$, $u_j^{(i)} \geq 0$ (unstable neurons). Activate and inactive neurons are already linear functions, so only unstable neurons require branching. In Section 3, we will show that after branching on some neurons, some previously unstable neurons become stable (active and inactive), and we will leverage these bound implications to avoid unnecessary branching on already-stable neurons, which results in a more efficient BaB process.

## 3 FINDING BOUND IMPLICATIONS FOR NN VERIFICATION

**Overview.** In this section, we introduce our optimization-based formulation and give an efficient procedure to quickly generate valid bound implications among all pairs of unstable neurons from *any* layers. We do this by reusing coefficients in bound propagation when intermediate layer bounds are computed before BaB, and solving simple linear programming (LP) problems efficiently without relying on an external solver. With the large number of implications found by our novel algorithm, we build a bound implication graph (BIG) (similar to the dependency graph in (Kouvaros & Lomuscio, 2021)) to accelerate the BaB process. We also demonstrate that multiple BIGs can be built for different subsets of subproblems during BaB, further improving its effectiveness.

**Bound implication: intuitions with linear bounds.** When assuming one neuron's bound to be within a smaller range (e.g., constraining an unstable neuron to be active or inactive), the bounds of another neuron can be conditionally tightened, and may become an active or inactive neuron. For example, considering two neurons at *any* two layers $i_1$, $i_2$ with pre-calculated linear inequality coefficients and biases by some popular verifiers like CROWN or $\alpha$-CROWN:

$$z_{j_1}^{(i_1)} \geq 2x + 1; \quad z_{j_2}^{(i_2)} \geq -x + 0.5$$

Given the one-dimensional input perturbation $-1 \leq x \leq 1$, both neurons are unstable since their bounds can either be positive or negative. However, if we consider the subproblem where $z_{j_1}^{(i_1)} \leq 0$, i.e., this neuron is split into the inactive case so consequentially $2x + 1 \leq 0$, then we know $x \leq -0.5$, hence the range of $z_{j_2}^{(i_2)}$ improves to $1 \leq z_{j_2}^{(i_2)} \leq 1.5$ accordingly, becoming an active neuron. In other words, if we split $z_{j_1}^{(i_1)}$ to be inactive, then $z_{j_2}^{(i_2)}$ will automatically become active, and such conditions commonly occur in BaB. In this example, the number of subproblems can be potentially reduced by $1/4$ (excluding one configuration of the four possibilities), saving BaB time.

**Finding bound implications: an optimization-based method.** A naive way to find the implied correlations between neurons is to directly apply a bound propagation method such as $\alpha$-CROWN: set a single unstable neuron's pre-activation bound $l_j^{(i)} = 0$ (active split) or $u_j^{(i)} = 0$ (inactive split), and update the bounds of *latter layers* via bound propagation. However, a major limitation of this approach is its scalability — given that there might be thousands of unstable neurons in a network, we need to run bound propagation thousands of times to find all correlations. Additionally, it only finds implications for neurons *after* layer $i$, and cannot find correlations in the same layer or prior layers.

To make the process of finding bound implications more powerful and efficient, we propose to reuse the intermediate linear bounds during bound propagation, namely the linear relationships in Eq. (3). Given two unstable neurons, $z_{i_2}^{(i_1)}$ and $z_{j_2}^{(i_2)}$, linear bound propagation such as $\alpha$-CROWN is used to compute their linear lower and upper bounds, denoted as $l_{j_1}^{(i_1)}, u_{j_1}^{(i_1)}, l_{j_2}^{(i_2)}, u_{j_2}^{(i_2)}$:

$$\min_{\boldsymbol{x} \in \mathcal{C}} z_{j_1}^{(i_1)} \geq \min_{\boldsymbol{x} \in \mathcal{C}} \underline{\mathbf{a}}^{(i_1,j_1)^\top} \boldsymbol{x} + \underline{c}^{(i_1,j_1)} := l_{j_1}^{(i_1)} \; ; \quad \min_{\boldsymbol{x} \in \mathcal{C}} z_{j_2}^{(i_2)} \geq \min_{\boldsymbol{x} \in \mathcal{C}} \underline{\mathbf{a}}^{(i_2,j_2)^\top} \boldsymbol{x} + \underline{c}^{(i_2,j_2)} := l_{j_2}^{(i_2)}$$

$$\min_{\boldsymbol{x} \in \mathcal{C}} (-z_{j_1}^{(i_1)}) \geq \min_{\boldsymbol{x} \in \mathcal{C}} \overline{\mathbf{a}}^{(i_1,j_1)^\top} \boldsymbol{x} + \overline{c}^{(i_1,j_1)} := u_{j_1}^{(i_1)}; \; \min_{\boldsymbol{x} \in \mathcal{C}} (-z_{j_2}^{(i_2)}) \geq \min_{\boldsymbol{x} \in \mathcal{C}} \overline{\mathbf{a}}^{(i_2,j_2)^\top} \boldsymbol{x} + \overline{c}^{(i_2,j_2)} := u_{j_2}^{(i_2)}$$

Here $\underline{\mathbf{a}}, \underline{c}$ and $\overline{\mathbf{a}}, \overline{c}$ are coefficients for linear lower and upper bounds, respectively. These linear bounds must *have already been computed* when calculating intermediate layer bounds $l_{j_1}^{(i_1)}, l_{j_2}^{(i_2)}$, $u_{j_1}^{(i_1)}, u_{j_2}^{(i_2)}$ because they are needed during the bounding step in BaB, and no extra cost is further needed. Due to symmetry, we focus on one case, $l_{j_1}^{(i_1)}$. When the input set $\mathcal{C}$ is a $\ell_\infty$-norm constraint, $l_{j_1}^{(i_1)}$ is computed by solving this linear programming problem:

$$l_{\text{no-imp}}^* := \min_{\boldsymbol{x}} \underline{\mathbf{a}}^{(i_1,j_1)^\top} \boldsymbol{x} + \underline{c}^{(i_1,j_1)} \quad \text{s.t. } \boldsymbol{x}_0 - \epsilon \leq \boldsymbol{x} \leq \boldsymbol{x}_0 + \epsilon \tag{4}$$

It has a closed-form solution in $O(d_0)$ time: $l_{j_1}^{(i_1)} := l_{\text{no-imp}}^* = -\epsilon \|\underline{\mathbf{a}}^{(i_1,j_1)}\|_1 + \underline{\mathbf{a}}^{(i_1,j_1)^\top} \boldsymbol{x}_0 + \underline{c}^{(i_1,j_1)}$. Now we consider the lower bound $l_{j_1}^{(i_1)}$ for the **implicated neuron** $z_{j_1}^{(i_1)}$, assuming that another **implicant neuron** $z_{j_2}^{(i_2)}$ is set to be inactive, i.e., $z_{j_2}^{(i_2)} \leq 0$. Under this assumption, the bound on $l_{j_1}^{(i_1)}$ can be potentially improved with the additional constraint $z_{j_2}^{(i_2)} \leq 0$:

$$l_{\text{imp}}^* := \min_{\boldsymbol{x}} \underline{\mathbf{a}}^{(i_1,j_1)^\top} \boldsymbol{x} + \underline{c}^{(i_1,j_1)} \quad \text{s.t. } \boldsymbol{x}_0 - \epsilon \leq \boldsymbol{x} \leq \boldsymbol{x}_0 + \epsilon;$$
$$z_{j_2}^{(i_2)} \leq 0 \quad \text{(implicant constraint)} \tag{5}$$

However, it is intractable to solve Eq. (5) since $z_{j_2}^{(i_2)}$ is generally an non-convex function of $\boldsymbol{x}$. Considering the set $S := \{\boldsymbol{x} : z_{j_2}^{(i_2)} \leq 0, \boldsymbol{x} \in \mathcal{C}\}$ is a subset of $\bar{S} := \{\boldsymbol{x} : \underline{\mathbf{a}}^{(i_2,j_2)^\top} \boldsymbol{x} + \underline{c}^{(i_2,j_2)} \leq 0, \boldsymbol{x} \in \mathcal{C}\}$, we can solve a relaxed LP with a larger feasible set $\bar{S}$ instead:

$$l_{\text{relaxed}}^* := \min_{\boldsymbol{x}} \underline{\mathbf{a}}^{(i_1,j_1)^\top} \boldsymbol{x} + \underline{c}^{(i_1,j_1)} \quad \text{s.t. } \boldsymbol{x}_0 - \epsilon \leq \boldsymbol{x} \leq \boldsymbol{x}_0 + \epsilon;$$
$$\underline{\mathbf{a}}^{(i_2,j_2)^\top} \boldsymbol{x} + \underline{c}^{(i_2,j_2)} \leq 0 \quad \text{(relaxed implicant constraint)} \tag{6}$$

This LP problem has a special structure that we can explicitly construct an algorithm to solve Eq. (6) in $O(d_0 \log d_0)$ time where $d_0$ is the dimension of $\boldsymbol{x}$, based on the fact that there is only one dual variable whose value can be sorted and searched. The algorithm is presented in Appendix C.1.

**Theorem 3.1.** *Given two unstable neurons and the LP formulations in Eq. (4), (5) and (6), the following holds: (1) The LP in Eq. (6) is always feasible; (2) $l_{\text{no-imp}}^* \leq l_{\text{relaxed}}^* \leq l_{\text{imp}}^*$.*

Theorem 3.1 shows that our tractable formulation Eq. (6) always produces a valid lower bound for the solution of the intractable Eq. (5), and the lower bound of the implicated neuron, $l_{j_1}^{(i_1)}$ can be potentially improved given the implicant $z_{j_2}^{(i_2)} \leq 0$. In other words, an implicant neuron $z_{j_2}^{(i_2)}$ split to *inactive* case implies an improved *lower* bound of an implicated neuron $z_{j_1}^{(i_1)}$ based on our analysis, which may change $z_{j_1}^{(i_1)}$ from an unstable neuron to an active neuron due to its improved (greater) lower bound. In fact, there are four possible ways that two neurons may interact (*inactive/active* implicant and the *lower/upper* bound of the implicated neuron), so four LPs like Eq. (6) can be constructed for each pair of neurons and we gave these cases in Appendix A.

**Criteria of neurons selection.** Note that the above procedure does not limit the choice of unstable neurons, and the implicant and implicated neurons can be from *any* two layers $j_1, j_2$. Since every pair of unstable neurons can be considered, the total number of possible bound implications is large. In this section, we discuss a simple filtering process that helps us find the strong bound implications while filtering out unuseful ones. We first present a theorem about the objective improvement in (6):

**Theorem 3.2.** *Given two unstable neurons and the LP formulations in Equation (4), (6), if the following condition on an implicated neuron $(i_1, j_1)$ and an implicant neuron $(i_2, j_2)$ holds:*

$$\mathbf{a}^{(i_2,j_2)^\top} \boldsymbol{x}^* + \underline{c}^{(i_2,j_2)} \leq 0 \tag{7}$$

*where each element of $\boldsymbol{x}^*$ is chosen as (here the subscript $k$ means the $k$-th element in a vector):*

$$x_k^* := \begin{cases} x_{0,k} - \epsilon \cdot \underline{a}_k^{(i_1,j_1)}, & \text{if } \underline{a}_k^{(i_1,j_1)} > 0 \\ x_{0,k} + \epsilon \cdot \underline{a}_k^{(i_1,j_1)}, & \text{if } \underline{a}_k^{(i_1,j_1)} < 0 \\ x_{0,k} - \epsilon \cdot \text{sign}(\underline{a}_k^{(i_2,j_2)}), & \text{if } \underline{a}_k^{(i_1,j_1)} = 0 \end{cases}$$

*then $l_{relaxed}^* = l_{no\text{-}imp}^*$, i.e., the relaxed implicant constraint brings no improvement in objective (6).*

The theorem constructs a solution $\boldsymbol{x}^*$ to the problem Eq. (4), and we check if this solution already satisfies the added implicant constraint in Eq. (6). Note the the case of $\underline{a}_k^{(i_1,j_1)} = 0$ is necessary, since due to the convolutional architecture it is indeed possible that many elements of $\mathbf{a}^{(i_1,j_1)}$ are exactly 0. When the objective does not improve, the relaxed implicant constraint is redundant with two cases:

**Corollary 3.3.** *When $l_{relaxed}^* = l_{no\text{-}imp}^*$ in Theorem 3.2, one of the two cases must happen:*

1. $\left|\mathbf{a}^{(i_2,j_2)^\top} \boldsymbol{x}_0 + \underline{c}^{(i_2,j_2)}\right| - \epsilon\|\mathbf{a}^{(i_2,j_2)}\|_1 > 0$, *i.e., the relaxed implicant constraint is too far away from $\boldsymbol{x}_0$ and does not intersect with the $\ell_\infty$-norm box $\mathcal{C}$ centered at $\boldsymbol{x}_0$ (Fig. 2(a));*

2. $\left|\mathbf{a}^{(i_2,j_2)^\top} \boldsymbol{x}_0 + \underline{c}^{(i_2,j_2)}\right| - \epsilon\|\mathbf{a}^{(i_2,j_2)}\|_1 \leq 0$ *but $\mathbf{a}^{(i_2,j_2)^\top} \boldsymbol{x}^* + \underline{c}^{(i_2,j_2)} \leq 0$, i.e., the constraint is not binding and does not change the solution of Eq. 4 (Fig. 2(b)).*

Based on Theorem 3.2 and Corollary 3.3, we use the following filtering procesure to quickly select pairs of neurons to find effective bound implications. We first select implicant neurons. Since the first case in Corollary 3.2 only depends on the implicant but not the implicated neuron, all unstable neurons whose linear bounds do not intersect with $\mathcal{C}$ are not useful as implicants and are removed. Then, we rank the remaining neurons by the distance of their implicant constraint to $\boldsymbol{x}_0$ from near to far, since a constraint

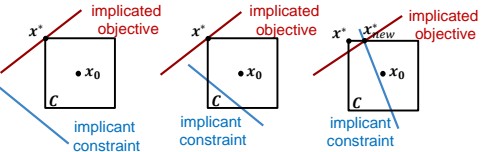

(a) No improvement  (b) No improvement  (c) Objective improved

Figure 2: Objective improvements depend on the intersection of implicant constraint with $\mathcal{C}$.

closer to $\boldsymbol{x}_0$ is more likely to cut off more feasible regions in Eq. (6). Based on the ranking, we select top-$K$ neurons as the implicant neurons. Then we apply Theorem 3.2 with all unstable neurons as the implicated neurons and the selected $K$ implicant neurons, and remove the implicated neurons whose bounds cannot be improved based on condition (7). Finally, we solve Eq. (6) using our fast $O(d_0 \log d_0)$ solver (Algorithm 1 in Appendix C) for all remaining effective pairs of implicant and implicated neurons. Its *complexity depends on the input dimension $d_0$ and $K$, without direct dependencies on DNN size*. The process is done on GPUs within a few seconds even for large DNNs.

**Construct the Bound Implication Graph (BIG).** After finding bound implications using our novel formulation above, we can generate a directed graph, the Bound Implication Graph (BIG), following a similar procedure of building a dependency graph in previous work (Kouvaros & Lomuscio, 2021). Here a BIG is a graph $\mathbf{G} = (\mathbf{V}, \mathbf{E})$, where $\mathbf{V} = \{P_j^{(i)} | i \in [L], j \in [d_i]\} \cup \{Q_j^{(i)} | i \in [L], j \in [d_i]\}$ is the node set with cardinality $|\mathbf{V}| = 2\sum_{i=1}^L d_i$; since each neuron can be split into either active or inactive cases, node $P_j^{(i)}$ and $Q_j^{(i)}$ reflects the two cases of the neuron $z_j^{(i)}$ (active and inactive, respectively). $\mathbf{E} \subseteq \{(R_{j_1}^{(i_1)}, R_{j_2}^{(i_2)}) | R \in \{P, Q\}, i_1, i_2 \in [L], j_1 \in [d_{i_1}], j_2 \in [d_{i_2}], (i_1, j_1) \neq (i_2, j_2)\}$. An edge $e = (P_{j_1}^{(i_1)}, Q_{j_2}^{(i_2)})$ indicates that the split $z_{j_1}^{(i_1)} \geq 0$ (active), implies $z_{j_2}^{(i_2)}$ is inactive. For example, after obtaining improved bounds for $z_{j_1}^{(i_1)}$ by implying $z_{j_2}^{(i_2)}$ is inactive ($z_{j_2}^{(i_2)} \leq 0$) solving (6), if the improved objective $l_{relaxed}^* \geq 0$, indicating $z_{j_1}^{(i_1)}$ becomes active, we add

edge $(Q_{j_2}^{(i_2)}, P_{j_1}^{(i_1)})$ to the graph. Note that in practice, we only need to maintain unstable neurons in this graph. In addition, the bound implications can be propagated through the graph until reaching nodes with zero out-degree, so although the bound implications found by solving (6) is only for a pair of neurons, more neurons can be connected on the graph (Figure 1c).

During BaB, when we split $z_j^{(i)}$ into the active (or inactive) case, we traverse BIG starting from the node $P_j^{(i)}$ (or $Q_j^{(i)}$) and find all connected nodes. If there is a path from $P_j^{(i)}$ to $P_{j'}^{(i')}$ (or $Q_{j'}^{(i')}$), we can add the additional split $z_{j'}^{(i')} >= 0$ (or $z_{j'}^{(i')} <= 0$) to this subproblem. We often find that splitting one single neuron may imply multiple other neurons are active or inactive. This greatly improves the effectiveness of BaB since these implied neurons will not need splits anymore, and these additional constraints help to obtain tighter bounds. We show our full algorithm to construct BIG and use BIG during branch and bound in Algorithm 3 and 4 in Appendix C repectively.

**Subproblem-specific Bound implications.** We propose one additional approach to enhance BIG. In BaB, many subproblems share the same splits constraints; for example, if the first neuron chosen by BaB is $z_{j_1}^{(i_1)}$, then the constraint $z_{j_1}^{(i_1)} \geq 0$ ($z_{j_1}^{(i_1)}$ is active) or $z_{j_1}^{(i_1)} \leq 0$ ($z_{j_1}^{(i_1)}$ is inactive) will appear in all subproblems created later. Considering the first $M$ split by BaB, up to $2^M$ initial subproblems are created, and all subsequently generated subproblems contain $M$ constraints that are the same as those in one of the initial subproblems. If we start the construction of BIG after the first $M$ splits, we can build up to $2^M$ different BIGs and each BIG may benefit from additional $M$ split constraints. In this case, each BIG is subproblem-specific (called "subBIG") because it includes constraints valid only in a specific subset of subproblems. To build subBIGs, we extend bound implications with multiple constraints from initial branches. We derive from the Eq. (5) with the implicant constraint and additional constraints:

$$l_{\text{relaxed-multi}}^* := \min_{\boldsymbol{x}} \ \underline{\mathbf{a}}^{(i_1, j_1)^\top} \boldsymbol{x} + \underline{c}^{(i_1, j_1)}$$

$$\text{s.t. } \boldsymbol{x}_0 - \epsilon \leq \boldsymbol{x} \leq \boldsymbol{x}_0 + \epsilon; \quad \underline{\mathbf{a}}^{(i_2, j_2)^\top} \boldsymbol{x} + \underline{c}^{(i_2, j_2)} \leq 0 \quad \text{(implicant constraint)} \qquad (8)$$

$$\underline{\mathbf{a}}^{(\boldsymbol{m}[k], \boldsymbol{n}[k])^\top} \boldsymbol{x} + \underline{c}^{(\boldsymbol{m}[k], \boldsymbol{n}[k])} \leq 0, \ k = 1, 2, 3, ..., M \quad \text{(additional split constraints)}$$

However, with constraints, $l_{\text{relaxed-multi}}^*$ has no closed-form solution. We construct Lagrangian multipliers and solve the dual problem by gradient descent as shown in Appendix D.

**Discussion.** Our framework of finding bound implications has several benefits. First, it does not rely on expensive solvers and reuses the linear bounds for computing intermediate layer bounds for free, which is very suitable for popular bound-propagation-based NN verifiers. Second, our bound implications can be found between two neurons from any layers (we have no restrictions on $j_1$, $j_2$ above), capturing both inter- and intra-layer neuron correlations. Third, our algorithm is scalable regarding DNN size because its complexity depends on the number of selected unstable neurons for finding the bound implications, and the optimization problem (5) does not depend on DNN size (except for the input dimension). Finally, BIG is independent of the actual BaB verifier used; although our evaluation uses $\beta$-CROWN as the base BaB verifier, it can also strengthen some new BaB verifiers not evaluated in this work. In Sec. 4, we show that our strong optimization-based formulation leads to finding significantly more neuron implications compared to previous works (Kouvaros & Lomuscio, 2021; Botoeva et al., 2020), and also noticeably improves state-of-the-art BaB verifiers.

## 4 EXPERIMENTS

We first compare our approach to existing work that also discover bound implications or dependencies among neurons. Then, we evaluate our bound implication graph (BIG)-enhanced verifier, on various verification benchmarks. We leave more details about experiment setup in Appendix E.

**Bound Implications Comparison.** We first evaluate the number of bound implications generated by Venus2 (Kouvaros & Lomuscio, 2021) and ours. We conduct experiments on MNIST-MLP and CIFAR-MLP used in (Botoeva et al., 2020), CIFAR-CNN-A-Adv and CIFAR-CNN-A-Mix from (Dathathri et al., 2020). We compare all implications generated before BaB and skip data points that can be verified by initial MIP or $\alpha$-CROWN since there is no necessity to use BaB. In Table 1, we show that our method performs significantly better than Venus2. Especially in CNN models, Venus2 cannot find intra-layer implications because it considers implications using intermediate layer bounds of one convolutional layer only, and no implications can be found because the convolutional layer

Table 1: Number of bound implications generated by different methods on 4 models. Following (Botoeva et al., 2020), we report inter-layer ($i_1 \neq i_2$) and intra-layer ($i_1 = i_2$) implications separately.

| | | MNIST-MLP | CIFAR-MLP | CIFAR-CNN-A-Adv | CIFAR-CNN-A-Mix |
|---|---|---|---|---|---|
| Avg. inter-layers implications | Venus2 | 56.15 | 4.66 | 6.42 | 19.17 |
| | Ours | **203.5** | **397.5** | **340.0** | **486.5** |
| Avg. intra-layers implications | Venus2 | 9.80 | 35.33 | 0 | 0 |
| | Ours | **263.4** | **502.1** | **763.4** | **803.2** |

weights are sparse and uncorrelated. In contrast, our optimization-based formulation found hundreds of implications that involve neurons from the same layer or different layers.

**A New Benchmark with Hard Verification instances.** Although we included several standard benchmarks used in previous papers showing consistent improvements, we also identified their shortcomings. First, many samples in these benchmarks can be verified by simple verification algorithms (such as CROWN/DeepPoly) or attacked via PGD attacks, so they cannot accurately reflect the recent progress of DNN verifiers. On the other hand, certain hard benchmarks have many unsolved instances with unknown difficulty, and it is unclear whether they can be actually solved. With only these benchmarks, it is difficult to gauge the advancement of verification tools.

Traditionally, the oval20 benchmark (Lu & Kumar, 2020; De Palma et al., 2021b) consists of hard instances that can be solved using BaBSR (Bunel et al., 2020) with a long timeout threshold. Since the difficulty of all benchmark instances is known with the groundtruth, many different tools are evaluated on this benchmark and it was easy to see the progress of verifiers. However, the latest verifiers can solve this benchmark within a few seconds per instance (Zhang et al., 2022). We thus propose a new benchmark **VeriHard** *in the same spirit*, which contains instances that can be solved using state-of-the-art verifiers with a long timeout. Unlike oval20, the benchmark consists of a mix of MNIST, CIFAR-10, CIFAR-100, and TinyImageNet models, including large ResNets. By comparing the performance of different verifiers under a shorter timeout, it can be easily used to gauge the improvement of future verifiers. We provide more details regarding this benchmark in Appendix E.1.

**Experimental setup.** Our implementation is based on the open-source $\alpha,\beta$-CROWN verifier, with our addition of BIG and subBIGs integrated. Since our method is *independent of the actual BaB solver* used, we use the popular $\beta$-CROWN algorithm as our baseline BaB method. Since new BaB methods may also benefit from our algorithm, the *improvements over the base BaB method* are the most important metric. For BIG and subBIGs construction, we select top-$K$ implicant neurons to derive our paired bound implications with $K = 1000$. For subBIGs, we use first $M = 8$ splits by BaB and apply a 50-steps gradient descent with Adam optimizer to optimize our objective $l^*_{\text{relaxed-multi}}$ on up to $2^M = 256$ unsolved subproblems. We leave more experimental details in Appendix E.2.

**Verification Results on VeriHard benchmarks.** We evaluate our BIG-enhanced verifiers, BIG and subBIGs, on new proposed **VeriHard** benchmark. Instances included in **VeriHard** benchmark are designed to be so challenging that state-of-the-art verification algorithms must make a good effort (long timeout threshold) to verify them. For comparison, we include multiple strong verification tools including PRIMA (Müller et al., 2022b), MN-BaB (Ferrari et al., 2021) and $\beta$-CROWN (Wang et al., 2021). From Table 2 we can see that, the percentage of verified instances is consistently lower than $50\%$ even for strong verifiers, and our BIG-enhanced verifier could solve 5 to 8 more instances compared to the strongest baseline we have on MNIST, CIFAR-10 benchmarks. For CIFAR-100 and TinyImageNet benchmarks, with large ResNet models, BIG-enhanced verifier could noticeably reduce the explored branches compared to the $\beta$-CROWN verifier as the baseline. Figure 3 shows the percentage of solved properties on **VeriHard** MNIST and CIFAR10 benchmarks vs. running time. Though all instances are hard enough and need to be verified through BaB with at least 150 seconds, BIG-enhanced verifier could consistently reduce the overall running time by reducing the overall branched neurons and branching iterations.

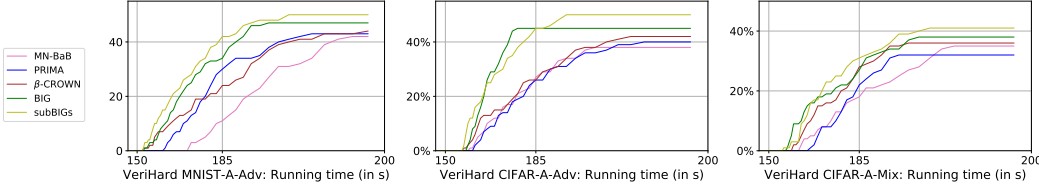

Figure 3: Percentage of solved properties on **VeriHard** {MNIST-A-Adv, CIFAR-A-Adv, CIFAR-A-Mix} benchmarks vs. time (timeout 200s). Our BIG and subBIGs consistently outperform baselines.

Table 2: Percentage of verified instances, average runtime, and the number of BaB branches (lower is better) on VeriHard benchmarks. We bold the highest verified percentage and the smallest number of branches. $\beta$-CROWN is used as our base BaB solver; BIG and subBIGs are added into $\beta$-CROWN.

| VeriHard Benchmarks | Metrics | CROWN/ DeepPoly | PRIMA | MN-BaB | $\beta$-CROWN | BIG (ours) | subBIGs (ours) |
|---|---|---|---|---|---|---|---|
| MNIST-CNN-A-Adv | Ver.% | 0.0 | 43.0 | 42.0 | 44.0 | 47.0 | **50.0** |
| | Branches | - | - | - | 60324.6 | 55210.4 | **53147.1** |
| | Time(s) | 0.0 | 182.4 | 178.5 | 176.2 | 184.6 | 186.2 |
| CIFAR10-CNN-A-Adv | Ver.% | 0.0 | 40.0 | 38.0 | 42.0 | 45.0 | **50.0** |
| | Branches | - | - | - | 61012.0 | 56254.9 | **54104.8** |
| | Time(s) | 0.0 | 176.4 | 174.2 | 176.4 | 181.2 | 183.4 |
| CIFAR10-CNN-A-Mix | Ver.% | 0.0 | 32.0 | 35.0 | 36.0 | 38.0 | **41.0** |
| | Branches | - | - | - | 69012.5 | 65302.1 | **63111.7** |
| | Time(s) | 0.0 | 179.2 | 178.4 | 182.4 | 185.5 | 190.5 |
| CIFAR100-Small | Ver.% | 0.0 | 24.0 | 25.0 | 29.0 | 30.0 | **32.0** |
| | Branches | - | - | - | 74113.5 | 71054.9 | **70006.6** |
| | Time(s) | 0.0 | 220.4 | 231.6 | 230.2 | 229.2 | 241.4 |
| CIFAR100-Medium | Ver.% | 0.0 | 27.0 | 27.0 | **30.0** | **30.0** | **30.0** |
| | Branches | - | - | - | 78856.2 | 75249.5 | **74124.9** |
| | Time(s) | 0.0 | 220.4 | 231.6 | 239.4 | 234.2 | 230.2 |
| CIFAR100-Large | Ver.% | 0.0 | 29.0 | 28.0 | 29.0 | 29.0 | **31.0** |
| | Branches | - | - | - | 74016.2 | 72012.5 | **70102.0** |
| | Time(s) | 0.0 | 234.2 | 229.5 | 230.4 | 235.2 | 246.5 |
| TinyImageNet-Medium | Ver.% | 0.0 | 27.0 | 27.0 | 32.0 | **33.0** | **33.0** |
| | Branches | - | - | - | 64253.9 | 61104.7 | **60082.5** |
| | Time(s) | 0.0 | 221.4 | 230.5 | 225.8 | 235.9 | 240.2 |

**Verification Results on Existing benchmarks.** We also evaluate BIG-enhanced verifiers on a few commonly used existing benchmarks, including MNIST benchmarks (CNN-A-Adv from SDP-FO (Dathathri et al., 2020) and ConvSmall from ERAN (Singh et al., 2019a) models) and CIFAR-10 benchmarks (CNN-A-Mix, CNN-A-Adv from SDP-FO and ConvSmall from ERAN models), and compare them with other strong verifier baselines. We also provide the verified accuracy upper bound derived by PGD attack as a reference. Results are shown in Table 3. We find that our method could consistently reduce the number of branches explored during BaB compared to the baseline $\beta$-CROWN method, and the percentage of verified instances also consistently improved.

Table 3: Percentage of verified instances, average runtime, and the number of BaB branches (lower is better) on existing benchmarks. We bold the highest verified percentage and the smallest number of branches. $\beta$-CROWN is used as our base BaB solver; BIG and subBIGs are added into $\beta$-CROWN.

| Existing Benchmarks | Metrics | CROWN/ DeepPoly | PRIMA | SDP-FO | MN-BaB | Venus2 | $\beta$-CROWN | BIG | subBIGs | Upper Bound |
|---|---|---|---|---|---|---|---|---|---|---|
| MNIST-CNN-A-Adv | Ver.% | 1.0 | 44.5 | 43.4 | - | 35.5 | 70.5 | 71.0 | **71.5** | 76.5 |
| | Branches | - | - | - | - | - | 4021.1 | 3530.2 | **2069.1** | - |
| | Time(s) | 0.1 | 135.9 | >20h | - | 148.4 | 21.0 | 23.7 | 29.3 | - |
| MNIST-ConvSmall | Ver.% | 15.8 | 59.8 | - | - | - | 72.7 | 72.8 | **73.2** | 73.2 |
| | Branches | - | - | - | - | - | 2739.4 | 2510.4 | **2112.2** | - |
| | Time(s) | 3.0 | 42.0 | - | - | - | 5.89 | 7.47 | 9.07 | - |
| CIFAR10-CNN-A-Adv | Ver.% | 21.5 | 41.5 | 39.6 | 42.5 | **47.5** | 45.0 | 45.5 | 46.0 | 65.0 |
| | Branches | - | - | - | - | - | 20462.6 | 19340.6 | **12671.2** | - |
| | Time(s) | 0.5 | 4.8 | >25h | 68.3 | 26.0 | 110.2 | 110.8 | 142.5 | - |
| CIFAR10-CNN-A-Mix | Ver.% | 23.5 | 37.5 | 39.6 | 35.0 | 33.5 | 41.5 | 41.5 | **42.5** | 53.0 |
| | Branches | - | - | - | - | - | 22344.4 | 21996.8 | **20462.9** | - |
| | Time(s) | 0.4 | 34.3 | >25h | 140.3 | 72.4 | 35.6 | 36.3 | 69.2 | - |
| CIFAR10-ConvSmall | Ver.% | 35.9 | 44.6 | - | - | - | 46.3 | 46.3 | **46.5** | 73.2 |
| | Branches | - | - | - | - | - | 5325.0 | 5072.0 | **4203.6** | - |
| | Time(s) | 4.0 | 13.0 | - | - | - | 5.25 | 5.85 | 7.02 | - |

**Impacts of BIG on branching.** In Table 4, we show the actual number of implicated neurons on BIG/subBIGs found by our algorithm in various networks, and the number is often large. In addition, we visualize how BIG helps branch and bound procedure in Figure 4, by showing the trends of 1) Global lower bounds; 2) Cumulative explored branches; 3) Used bound implications vs. the branching iterations. We compare our BIG-enhanced verifier (BIG, subBIGs) to the baseline $\beta$-CROWN verifier. We can find that both BIG and subBIGs could steadily tighten the lower bound during BaB, and noticeably reduce the number of explored branches towards complete verification. In addition, during each branching round, we can fix several additional *implicant neurons* from unstable to stable to tighten the intermediate bounds thus global lower bounds. We leave more visualization results in Appendix F.

Table 4: The number of implicated neurons found in benchmarks of **VeriHard** dataset.

| VeriHard Benchmark | Bound Implications | |
|---|---|---|
| | BIG | subBIGs |
| MNIST-A-Adv | 934.25 | 1424.48 |
| CIFAR10-A-Adv | 5603.46 | 8403.24 |
| CIFAR100-Medium | 16624.05 | 20042.23 |
| TinyImageNet-Medium | 15352.49 | 18023.23 |

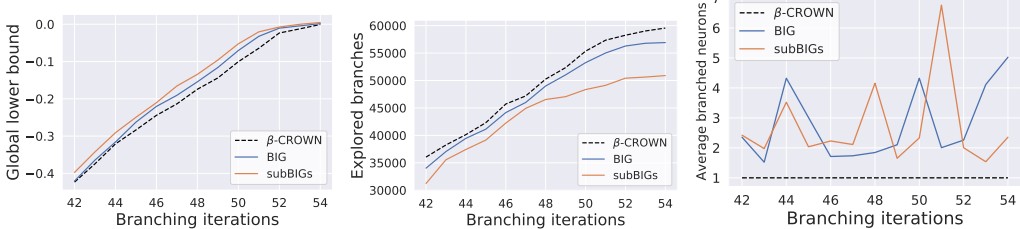

Figure 4: {(**Left**) Global lower bounds; (**Middle**) Total explored branches; (**Right**) Average used bound implications} trends along with branching iterations on one specific instance in **VeriHard**-CIFAR10-A-Adv benchmark. More than one neurons are branched per iteration thanks to BIG, and the bounds and the number explored branches both improve noticeably. Baseline BaB methods like $\beta$-CROWN can branch only one neuron per BaB iteration.

## 5 RELATED WORK

The DNN verification methods are divided into two categories, complete (Katz et al., 2017; Tjeng et al., 2019; Xu et al., 2020b; Wang et al., 2021; Bunel et al., 2018) and incomplete (Zhang et al., 2018; Wong & Kolter, 2018; Wang et al., 2018a;b). Complete verification is proven NP-complete (Katz et al., 2017), and current verifiers are based on SMT (Katz et al., 2017), off-shelf solvers (Tjeng et al., 2019; Bunel et al., 2018; Dutta et al., 2017), and specialized branch-and-bound (BaB) (Bunel et al., 2018; Wang et al., 2021; Bunel et al., 2020). Branch-and-bound is usually the most effective technique in practice and has been used in top toolkits in the verification of neural network competitions (Bak et al., 2021; Müller et al., 2022a). On the other hand, incomplete verification is usually efficient with polynomial time complexity but is conservative. Existing incomplete verifiers are based on bound propagation (Wang et al., 2018a; Zhang et al., 2018; Wang et al., 2018b; Xu et al., 2020a; Gowal et al., 2019), abstract interpretation (Singh et al., 2019b; Gehr et al., 2018; Müller et al., 2022b; Singh et al., 2019a), convex relaxation (Palma et al., 2021; Tjandraatmadja et al., 2020), duality (Wong & Kolter, 2018; De Palma et al., 2021b; Dvijotham et al., 2019), and semi-definite programming (Chiu & Zhang, 2023; Anderson et al., 2020a; Dathathri et al., 2020; Batten et al., 2021; Raghunathan et al., 2018). Specifically, the cheap bound propagation technique can be efficiently combined with the branch-and-bound to accelerate complete verification (Xu et al., 2020b; Wang et al., 2021).

Neuron dependency as a potential way to improve the tightness of bounds is also studied in a few papers (Anderson et al., 2020b; Palma et al., 2021; Müller et al., 2022b; Zhang et al., 2022), but they usually aim to improve the bounding step, and the implication relationships among ReLUs during branching were not discussed. Ehlers (2017) first discussed case-analysis-based search by recording the dependencies between ReLU splits to reduce the number of branches. The most relevant existing work is Venus and Venus2 (Botoeva et al., 2020; Kouvaros & Lomuscio, 2021), which also explores the dependency among unstable ReLU neurons, but with a weaker method to find neuron implications. First, for dependency from different layers (inter-layer), they only consider the case where a neuron in one layer is set to be *inactive* (0), and propagate the interval bounds to the layers following it; thus, only two out of four kinds of implications can be found, and the interval bounds can also too loose to find implications. Second, our method can find implications for neurons *before* the implicated neuron. For example, by setting a neuron in layer 3 to be active or inactive, we may imply that a neuron in layer 1 or 2 is active or inactive, impossible in the prior work. Third, for same-layer dependency, our methods use bound equations that are propagated to the input layer, which utilizes tighter bounds on the input $x_0$ rather than the relatively looser intermediate layer bounds. Finally, our formulation is optimization-based, allowing us to optimize and find the strongest implications. These benefits lead to significant improvements in finding neuron implications (even without considering subBIGs), as we demonstrated in Sec. 4.

## 6 CONCLUSION

In this paper, we proposed a novel technique to quickly find bound implications for branch-and-bound-based neural network verification algorithms. Our optimization-based formulation is less restrictive and stronger than prior work, allowing us to find more implications to guide the branch-and-bound process. Our method has been demonstrated on multiple benchmarks with consistent improvements in terms of verified accuracy and the number of branches.

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

# APPENDIX

## A    FOUR CASES OF NEURON INTERACTIONS

In Section 3, we discussed how to find neuron implications between two neurons. Since each ReLU neuron has two possible statuses (active or inactive), there are four kinds of possible implications. We list all four possibilities and their corresponding optimization formulation here:

1. An implicant neuron $z_{j_2}^{(i_2)}$ split to *inactive* case implies an improved *lower* bound of an implicated neuron $z_{j_1}^{(i_1)}$ (this is the case in (6) discussed in Section 3):

$$l_{\text{relaxed}}^* := \min_{\boldsymbol{x}} \underline{\mathbf{a}}^{(i_1,j_1)^\top} \boldsymbol{x} + \underline{c}^{(i_1,j_1)}$$

$$\text{s.t. } \boldsymbol{x}_0 - \epsilon \le \boldsymbol{x} \le \boldsymbol{x}_0 + \epsilon; \quad \underline{\mathbf{a}}^{(i_2,j_2)^\top} \boldsymbol{x} + \underline{c}^{(i_2,j_2)} \le 0$$

(9)

2. An implicant neuron $z_{j_2}^{(i_2)}$ split to *active* case implies an improved *lower* bound of an implicated neuron $z_{j_1}^{(i_1)}$:

$$l_{\text{relaxed}}^* := \min_{\boldsymbol{x}} \underline{\mathbf{a}}^{(i_1,j_1)^\top} \boldsymbol{x} + \underline{c}^{(i_1,j_1)}$$

$$\text{s.t. } \boldsymbol{x}_0 - \epsilon \le \boldsymbol{x} \le \boldsymbol{x}_0 + \epsilon; \quad \overline{\mathbf{a}}^{(i_2,j_2)^\top} \boldsymbol{x} + \overline{c}^{(i_2,j_2)} \ge 0$$

(10)

3. An implicant neuron $z_{j_2}^{(i_2)}$ split to *inactive* case implies an improved *upper* bound of an implicated neuron $z_{j_1}^{(i_1)}$:

$$l_{\text{relaxed}}^* := \max_{\boldsymbol{x}} \overline{\mathbf{a}}^{(i_1,j_1)^\top} \boldsymbol{x} + \overline{c}^{(i_1,j_1)}$$

$$\text{s.t. } \boldsymbol{x}_0 - \epsilon \le \boldsymbol{x} \le \boldsymbol{x}_0 + \epsilon; \quad \underline{\mathbf{a}}^{(i_2,j_2)^\top} \boldsymbol{x} + \underline{c}^{(i_2,j_2)} \le 0$$

(11)

4. An implicant neuron $z_{j_2}^{(i_2)}$ split to *active* case implies an improved *upper* bound of an implicated neuron $z_{j_1}^{(i_1)}$:

$$l_{\text{relaxed}}^* := \max_{\boldsymbol{x}} \overline{\mathbf{a}}^{(i_1,j_1)^\top} \boldsymbol{x} + \overline{c}^{(i_1,j_1)}$$

$$\text{s.t. } \boldsymbol{x}_0 - \epsilon \le \boldsymbol{x} \le \boldsymbol{x}_0 + \epsilon; \quad \overline{\mathbf{a}}^{(i_2,j_2)^\top} \boldsymbol{x} + \overline{c}^{(i_2,j_2)} \ge 0$$

(12)

Note that all four optimization problems have the same nature and can be solved using the same technique discussed in Section C.1. we will only use Eq. (6) as an example since all other three cases can be converted to Eq. (6). For instance, to solve Eq. (10), we can change the implicant constraint to $-\underline{\mathbf{a}}^{(i_2,j_2)^\top} \boldsymbol{x} - \underline{c}^{(i_2,j_2)} \le 0$; to solve Eq. (11), we can change the $\max$ objective to $\min_{\boldsymbol{x}} -\underline{\mathbf{a}}^{(i_1,j_1)^\top} \boldsymbol{x} - \underline{c}^{(i_1,j_1)}$.

## B    PROOFS

We now give the proofs for the two theorems in Section 3. Here Theorem 3.1 shows the possibility of solving a cheap linear programming problem to find bound implications, and Theorem 3.2 shows how we can reduce the number of optimization problems by filtering out some neurons that do not help find bound implications.

**Theorem 3.1.** *Given two unstable neurons and the LP formulations in Eq. (4), (5) and (6), the following holds: (1) The LP in Eq. (6) is always feasible; (2) $l_{\text{no-imp}}^* \le l_{\text{relaxed}}^* \le l_{\text{imp}}^*$.*

*Proof.* (1) To show that (6) is always feasible, we must show that there exists some $\boldsymbol{x}$ such that $\boldsymbol{x}_0 - \epsilon \le \boldsymbol{x} \le \boldsymbol{x}_0 + \epsilon$ also satisfies the other constraint $\underline{\mathbf{a}}^{(i_2,j_2)^\top} \boldsymbol{x} + \underline{c}^{(i_2,j_2)} \le 0$.

Given that the neuron $z_{j_2}^{(i_2)}$ is an unstable neuron, we know that $l_{j_2}^{(i_2)} \le 0$. By definition of $l_{j_2}^{(i_2)}$, we know that $\min_{\boldsymbol{x}_0 - \epsilon \le \boldsymbol{x} \le \boldsymbol{x}_0 + \epsilon} \underline{\mathbf{a}}^{(i_2,j_2)^\top} \boldsymbol{x} + \underline{c}^{(i_2,j_2)} \le 0$, so such a $\boldsymbol{x}$ must exist to satisfy the constraint.

(2) The objective $l^*_{\text{relaxed}}$ and $l^*_{\text{imp}}$ have one additional constraint compared to $l^*_{\text{no-imp}}$, so $l^*_{\text{no-imp}} \leq l^*_{\text{relaxed}}$ and $l^*_{\text{no-imp}} \leq l^*_{\text{imp}}$. It remains to prove $l^*_{\text{relaxed}} \leq l^*_{\text{imp}}$.

Define $S := \{\boldsymbol{x} : z^{(i_2)}_{j_2}(\boldsymbol{x}) \leq 0, \boldsymbol{x} \in \mathcal{C}\}$ and $\bar{S} := \{\boldsymbol{x} : \underline{\mathbf{a}}^{(i_2,j_2)\top}\boldsymbol{x} + \underline{c}^{(i_2,j_2)} \leq 0, \boldsymbol{x} \in \mathcal{C}\}$. Here $\mathcal{C}$ is the set $\boldsymbol{x}_0 - \epsilon \leq \boldsymbol{x} \leq \boldsymbol{x}_0 + \epsilon$. We claim that $S \subseteq \bar{S}$. This is true because given any $\boldsymbol{x}' \in \mathcal{C}$, if $z^{(i_2)}_{j_2}(\boldsymbol{x}') \leq 0$ (by definition, $\boldsymbol{x}' \in S$), we have $\underline{\mathbf{a}}^{(i_2,j_2)\top}\boldsymbol{x}' + \underline{c}^{(i_2,j_2)} \leq z^{(i_2)}_{j_2}(\boldsymbol{x}') \leq 0$ since $\underline{\mathbf{a}}^{(i_2,j_2)\top}\boldsymbol{x}' + \underline{c}^{(i_2,j_2)}$ is a linear lower bound of $z^{(i_2)}_{j_2}(\boldsymbol{x}')$ found by $\alpha$-CROWN. Thus, $S \subseteq \bar{S}$.

Since $S \subseteq \bar{S}$, (6) has the equal or larger feasible region compared to (5), and thus $l^*_{\text{relaxed}} \leq l^*_{\text{imp}}$. $\square$

**Theorem 3.2.** *Given two unstable neurons and the LP formulations in (4), (6), if the following condition holds:*
$$\underline{\mathbf{a}}^{(i_2,j_2)\top}\boldsymbol{x}^* + \underline{c}^{(i_2,j_2)} \leq 0$$
*where each element of $\boldsymbol{x}^*$ is chosen as (here the subscript $k$ means the $k$-th element in a vector):*
$$x^*_k := \begin{cases} x_{0,k} - \epsilon \cdot \underline{a}^{(i_1,j_1)}_k, & \text{if } \underline{a}^{(i_1,j_1)}_k > 0 \\ x_{0,k} + \epsilon \cdot \underline{a}^{(i_1,j_1)}_k, & \text{if } \underline{a}^{(i_1,j_1)}_k < 0 \\ x_{0,k} - \epsilon \cdot \text{sign}(\underline{a}^{(i_2,j_2)}_k), & \text{if } \underline{a}^{(i_1,j_1)}_k = 0 \end{cases}$$
*then $l^*_{relaxed} = l^*_{no\text{-}imp}$, i.e., there is no improvement. Additionally, one of the two cases must happen:*

1. $\left| \underline{\mathbf{a}}^{(i_2,j_2)\top}\boldsymbol{x}_0 + \underline{c}^{(i_2,j_2)} \right| - \epsilon\|\underline{\mathbf{a}}^{(i_2,j_2)}\|_1 > 0$, *i.e., the relaxed implicant constraint does not have an intersection with the $\ell_\infty$ box $\mathcal{C}$;*

2. $\left| \underline{\mathbf{a}}^{(i_2,j_2)\top}\boldsymbol{x}_0 + \underline{c}^{(i_2,j_2)} \right| - \epsilon\|\underline{\mathbf{a}}^{(i_2,j_2)}\|_1 \leq 0$ *but* $\underline{\mathbf{a}}^{(i_2,j_2)\top}\boldsymbol{x}^* + \underline{c}^{(i_2,j_2)} \leq 0$.

*Proof.* The linear programming problem (4) has a closed form optimal solution:
$$x'_k := \begin{cases} x_{0,k} - \epsilon \cdot \underline{a}^{(i_1,j_1)}_k, & \text{if } \underline{a}^{(i_1,j_1)}_k > 0 \\ x_{0,k} + \epsilon \cdot \underline{a}^{(i_1,j_1)}_k, & \text{if } \underline{a}^{(i_1,j_1)}_k < 0 \\ \text{don't care}, & \text{if } \underline{a}^{(i_1,j_1)}_k = 0 \end{cases}$$
If this solution $x'_k$ already satisfies the additional constraint $\underline{\mathbf{a}}^{(i_2,j_2)\top}\boldsymbol{x} + \underline{c}^{(i_2,j_2)} \leq 0$ added in (6), this constraint is redundant and thus $l^*_{\text{relaxed}} = l^*_{\text{no-imp}}$.

For the indices $k$ where $\underline{a}^{(i_1,j_1)}_k = 0$, since $x'_k$ is unconstrained, we can choose $x'_k$ that minimize $\underline{\mathbf{a}}^{(i_2,j_2)\top}\boldsymbol{x} + \underline{c}^{(i_2,j_2)}$ to satisfy the new constraint in (6) as much as possible, by setting $x'_k = x_{0,k} - \epsilon \cdot \text{sign}(\underline{a}^{(i_2,j_2)}_k)$. So the setting of $x^*_k$ gives an optimal solution of (4) that has the minimum violation of the new constraint in (6). If $x^*_k$ satisfies the new implicant constraint, the constraint is redundant, and $l^*_{\text{relaxed}} = l^*_{\text{no-imp}}$.

Now, we define the set $\hat{S} := \{\boldsymbol{x} : \underline{\mathbf{a}}^{(i_2,j_2)\top}\boldsymbol{x} + \underline{c}^{(i_2,j_2)} \leq 0\}$. Due to Theorem 3.1, the LP in (6) is always feasible, so $\hat{S} \cap \mathcal{C} \neq \emptyset$. Geometrically, the constraint $\underline{\mathbf{a}}^{(i_2,j_2)\top}\boldsymbol{x} + \underline{c}^{(i_2,j_2)} \leq 0$ is redundant in two cases:

1. $\mathcal{C} \in \hat{S}$, or the $\ell_\infty$ box $\mathcal{C}$ does not intersect with the line $\underline{\mathbf{a}}^{(i_2,j_2)\top}\boldsymbol{x} + \underline{c}^{(i_2,j_2)} = 0$. In this case, the $\ell_\infty$ distance from this line to the origin is greater than $\epsilon$:

$$\frac{\left| \underline{\mathbf{a}}^{(i_2,j_2)\top}\boldsymbol{x}_0 + \underline{c}^{(i_2,j_2)} \right|}{\|\underline{\mathbf{a}}^{(i_2,j_2)}\|_1} > \epsilon$$

And this is the first case in this Theorem.

2. $\mathcal{C} \cap \hat{S} \neq \mathcal{C}$, when the line $\underline{\mathbf{a}}^{(i_2,j_2)\top}\boldsymbol{x} + \underline{c}^{(i_2,j_2)} = 0$ cuts through $\mathcal{C}$ ($\ell_\infty$ distance is less than or equal to $\epsilon$), however it does not remove the existing optimal solution for (4), i.e., $\underline{\mathbf{a}}^{(i_2,j_2)\top}\boldsymbol{x}^* + \underline{c}^{(i_2,j_2)} \leq 0$.

This theorem allows us to filter out implicant neurons that have large $\ell_\infty$ distance to the origin first. Using the remaining implicant neurons, we further check each implicated unstable ReLU neuron

to see if $x^x$ satisfies the above constraint. This allows us to eliminate these pairs of neurons in calculation.

$\square$

## C ALGORITHMS

We show the full algorithms discussed in Section 3 in this section.

### C.1 CLOSED-FORM SOLUTION OF THE RELAXED LP

In Section 3, we show that it is possible to use a fast optimization algorithm in $O(d_0 \log d_0)$ time, where $d_0$ is the neural network input dimension. Here we present this algorithm.

We solve the problem by adding a Lagrange multiplier $\rho$ and derive a dual form of Eq. (6):

$$
\begin{aligned}
&\min_{\boldsymbol{x}_0 - \epsilon \leq \boldsymbol{x} \leq \boldsymbol{x}_0 + \epsilon} \max_{\rho \geq 0} \ \underline{\mathbf{a}}^{(i_1,j_1)^\top} \boldsymbol{x} + \underline{c}^{(i_1,j_1)} + \rho(\underline{\mathbf{a}}^{(i_2,j_2)^\top} \boldsymbol{x} + \underline{c}^{(i_2,j_2)}) \\
&\geq \max_{\rho \geq 0} \min_{\boldsymbol{x}_0 - \epsilon \leq \boldsymbol{x} \leq \boldsymbol{x}_0 + \epsilon} (\underline{\mathbf{a}}^{(i_1,j_1)} + \rho \underline{\mathbf{a}}^{(i_2,j_2)})^\top \boldsymbol{x} + \underline{c}^{(i_1,j_1)} + \rho \underline{c}^{(i_2,j_2)} \\
&= \max_{\rho \geq 0} (\underline{\mathbf{a}}^{(i_1,j_1)} + \rho \underline{\mathbf{a}}^{(i_2,j_2)})^\top \boldsymbol{x}_0 + \underline{c}^{(i_1,j_1)} + \rho \underline{c}^{(i_2,j_2)} - \|(\underline{\mathbf{a}}^{(i_1,j_1)} + \rho \underline{\mathbf{a}}^{(i_2,j_2)})\|_1 \cdot \epsilon \\
&= \max_{\rho \geq 0} -\|(\underline{\mathbf{a}}^{(i_1,j_1)} + \rho \underline{\mathbf{a}}^{(i_2,j_2)})\|_1 \cdot \epsilon + \rho(\underline{\mathbf{a}}^{(i_2,j_2)^\top} \boldsymbol{x}_0 + \underline{c}^{(i_2,j_2)}) + \underline{\mathbf{a}}^{(i_1,j_1)} \boldsymbol{x}_0 + \underline{c}^{(i_1,j_1)}
\end{aligned}
\tag{13}
$$

Note that the inner minimization is solved in closed form using Hölder's inequality. The maximization problem over the dual variable $\rho$ is a one-dimensional, non-smooth, piece-wise linear, and concave optimization problem and can be solved by checking super-gradients at the endpoints of all linear pieces. We list the solving procedure in Algorithm 1:

---

**Algorithm 1** The closed-form solution of Eq. (6).

---

1: **Inputs**: $\underline{\mathbf{a}}^{(i_1,j_1)}, \underline{c}^{(i_1,j_1)}, \underline{\mathbf{a}}^{(i_2,j_2)}, \underline{c}^{(i_2,j_2)}, \boldsymbol{x}_0, \epsilon$

2: **Outputs**: The optimal solution of Eq. (6): $l^*_{\text{relaxed}}$

3: $\boldsymbol{q} \leftarrow -\underline{\mathbf{a}}^{(i_1,j_1)} / \underline{\mathbf{a}}^{(i_2,j_2)}$

4: $\boldsymbol{I} \leftarrow \text{argsort}(\boldsymbol{q})$       $\triangleright$ Dominates time complexity

5: $\underline{\mathbf{a}}^{(i_2,j_2)}_{\text{sorted}} \leftarrow \{\underline{\mathbf{a}}^{(i_2,j_2)}_{\boldsymbol{I}_1} \cdot \epsilon, \underline{\mathbf{a}}^{(i_2,j_2)}_{\boldsymbol{I}_2} \cdot \epsilon, ...., \underline{\mathbf{a}}^{(i_2,j_2)}_{\boldsymbol{I}_{d_0}} \cdot \epsilon\}$       $\triangleright$ Sort $\underline{\mathbf{a}}^{(i_2,j_2)}$ by index $I$ and scale by $\epsilon$

6: $\left(\underline{\mathbf{a}}^{(i_2,j_2)}_-\right)_i \leftarrow -\sum_{k=1}^{i}(|\underline{\mathbf{a}}^{(i_2,j_2)}_{\text{sorted}}|_k), \ i \in [d_0]$       $\triangleright$ Cumulative sum of $|\underline{\mathbf{a}}^{(i_2,j_2)}_{\text{sorted}}|$ by its length $d_0$

7: $\underline{\mathbf{a}}^{(i_2,j_2)}_+ \leftarrow \underline{\mathbf{a}}^{(i_2,j_2)}_- - \underline{\mathbf{a}}^{(i_2,j_2)}_{-d_0}$       $\triangleright$ Shift $\underline{\mathbf{a}}^{(i_2,j_2)}_-$ to positive range

8: $\nabla \underline{\mathbf{a}} \leftarrow \underline{\mathbf{a}}^{(i_2,j_2)}_+ + \underline{\mathbf{a}}^{(i_2,j_2)}_- + (\underline{\mathbf{a}}^{(i_2,j_2)^\top} \boldsymbol{x}_0 + \underline{c}^{(i_2,j_2)})$       $\triangleright$ Calculate super-gradient

9: $i^* \leftarrow i$ where $\nabla \underline{\mathbf{a}}_i = 0$       $\triangleright$ Find the index of super-gradient is 0

10: $\rho^* \leftarrow \max(\boldsymbol{q}_{\boldsymbol{I}_{i^*}}, 0)$       $\triangleright$ Find the best $\rho$ and introduce it to Eq. 13

11: $l^*_{\text{relaxed}} \leftarrow -\|(\underline{\mathbf{a}}^{(i_1,j_1)} + \rho^* \underline{\mathbf{a}}^{(i_2,j_2)})\|_1 \cdot \epsilon + \rho^*(\underline{\mathbf{a}}^{(i_2,j_2)^\top} \boldsymbol{x}_0 + \underline{c}^{(i_2,j_2)}) + \underline{\mathbf{a}}^{(i_1,j_1)} \boldsymbol{x}_0 + \underline{c}^{(i_1,j_1)}$

12: $l^*_0 \leftarrow -\|\underline{\mathbf{a}}^{(i_1,j_1)}\|_1 \cdot \epsilon + \underline{c}^{(i_2,j_2)} + \underline{\mathbf{a}}^{(i_1,j_1)} \boldsymbol{x}_0 + \underline{c}^{(i_1,j_1)}$       $\triangleright$ Objective when $\rho = 0$

13: **if** $l^*_{\text{relaxed}} < l^*_0$ **then**

14:      $l^*_{\text{relaxed}} \leftarrow l^*_0$       $\triangleright$ Compare to $l^*_0$, which is an additional end point

15: **Return:** $l^*_{\text{relaxed}}$

---

### C.2 ALGORITHM OF BIG CONSTRUCTION AND UTILIZATION

In Algorithm 3, We will introduce how to construct BIG after we obtain the linear equation of all unstable neurons (can be calculated by CROWN or $\alpha$-CROWN) in the verification process. Note that there are **For** loops Algorithm 3, the calculations inside them are independent and can be calculated in parallel though. In our experiments, we construct BIG efficiently by computing all implications only in four batches for four cases of neuron interactions.

---

**Algorithm 2** Filter Top-$K$ constraints by distance to $\boldsymbol{x}_0$.

---

1: **Inputs**: constraints of all unstable neurons $\underline{\mathbf{a}}, \overline{\mathbf{a}} \in \mathbb{R}^{N \times d_0}$ and $\underline{c}, \overline{c} \in \mathbb{R}^N$, where $N$ is number of
   unstable neurons and $d_0$ is the length of the model input $\boldsymbol{x}_0$; perturbation size $\epsilon$, $K$, $\underline{\mathbf{S}} = \emptyset$, $\overline{\mathbf{S}} = \emptyset$
2: **function** TOPKFILTERING($\underline{\mathbf{a}}, \overline{\mathbf{a}}, \underline{c}, \overline{c}, \boldsymbol{x}_0, \epsilon, K$)
3:     **for** $p = 1$ to $N$ **do**
4:         $\underline{\mathbf{S}} \cup |\underline{\mathbf{a}}^{(i_p, j_p)^\top} \boldsymbol{x}_0 + \underline{c}^{(i_p, j_p)}| - \epsilon \cdot ||\underline{\mathbf{a}}^{(i_p, j_p)}||_1$
5:         $\overline{\mathbf{S}} \cup |\overline{\mathbf{a}}^{(i_p, j_p)^\top} \boldsymbol{x}_0 + \overline{c}^{(i_p, j_p)}| - \epsilon \cdot ||\overline{\mathbf{a}}^{(i_p, j_p)}||_1$
6:     ▷ Sort from nearest to farthest distance
7:     $\underline{\mathbf{S}} = \texttt{argsort}(\underline{d}_p)[: K]$                  ▷ Indices of Top-$K$ inactive implicant constraints
8:     $\overline{\mathbf{S}} = \texttt{argsort}(\overline{d}_p)[: K]$                  ▷ Indices of Top-$K$ active implicant constraints
9: **return** $\underline{\mathbf{S}}, \overline{\mathbf{S}}$

---

## D    BOUND IMPLICATIONS WITH ADDITIONAL SPLIT CONSTRAINTS

To solve Eq. (8), we leverage $k$ Lagrange multipliers $\boldsymbol{\rho}_i$, where $i \in [1, 2, .., k]$ to derive the dual form solution:

$$\max_{\boldsymbol{\rho} \geq 0} \min_{\boldsymbol{x}} \underline{\mathbf{a}}^{(i_1, j_1)^\top} \boldsymbol{x} + \underline{c}^{(i_1, j_1)} + \sum_{k=1}^{M} \boldsymbol{\rho}_i (\underline{\mathbf{a}}^{(\boldsymbol{m}[k], \boldsymbol{n}[k])^\top} \boldsymbol{x} + \underline{c}^{(\boldsymbol{m}[k], \boldsymbol{n}[k])})$$
$$\text{s.t. } \boldsymbol{x}_0 - \epsilon \leq \boldsymbol{x} \leq \boldsymbol{x}_0 + \epsilon \tag{14}$$

For the $\ell_\infty$-norm constraint of $\boldsymbol{x}$, the inner minimization has a closed-form solution:

$$\max_{\boldsymbol{\rho} \geq 0} \underline{\mathbf{a}}^{(i_1, j_1)^\top} \boldsymbol{x}_0 + \underline{c}^{(i_1, j_1)} - \|\underline{\mathbf{a}}^{(i_1, j_1)}\|_1 \cdot \epsilon +$$
$$\sum_{k=1}^{M} \boldsymbol{\rho}_i \underline{\mathbf{a}}^{(\boldsymbol{m}[k], \boldsymbol{n}[k])^\top} \boldsymbol{x}_0 + \boldsymbol{\rho}_i \underline{c}^{(\boldsymbol{m}[k], \boldsymbol{n}[k])} - \|\boldsymbol{\rho}_i \underline{\mathbf{a}}^{(\boldsymbol{m}[k], \boldsymbol{n}[k])}\|_1 \cdot \epsilon \tag{15}$$

Then, we can easily solve $\boldsymbol{\rho}$ by projected gradient descent using an optimizer like Adam (Diederik et al., 2014).

## E    EXPERIMENTAL DETAILS

### E.1    VERIHARD BENCHMARK

The **VeriHard** benchmark includes models from existing benchmarks, such as MNIST-A-Adv, CIFAR-A-Adv, CIFAR-A-Mix from SDP-FO (Dathathri et al., 2020), and CIFAR100-small, CIFAR100-medium, CIFAR100-large, and TinyImageNet-medium from the neural networks competition (VNN-COMP (Bak et al., 2021; Müller et al., 2022a)). **VeriHard** applies a careful selection of perturbation size $\epsilon$ for each instance, aiming to increase the verification challenge while ensuring solvability. We will illustrate our $\epsilon$ configuration algorithm as follows:

We start by determining the unknown range $[\epsilon_l, \epsilon_r]$ of each instance. Given an instance with an initial $\epsilon_l = 0$, we increment $\epsilon_l$ step by step with step-size $\alpha = 0.01$, applying CROWN verification after each step, until the vanilla CROWN verifier fails to verify the instance at the current $\epsilon_l$. Conversely, we initialize $\epsilon_r$ to 1.0, decreasing it by the same step size 0.01 until PGD attack succeed with perturbation bound $\epsilon_r$. In this case, any $\epsilon$ configuration within the range $[\epsilon_l, \epsilon_r]$ enforces verification using the BaB process.

For determining the final $\epsilon$, we employ $\beta$-CROWN as our reference verifier. Binary search on $\epsilon$ is used to assess the feasibility of verifying the instance within a timeout $t$, which is randomly sampled from the interval $[0.8T, 1.5T]$ ($T$ is the final timeout designated for the instance). The resulting $\epsilon$ will make $\beta$-CROWN verifier running time to be as close to the given timeout $t$ as possible.

In **VeriHard** dataset, for MNIST and CIFAR10 instances, the actual timeout $T$ is set to 200 seconds, whereas for CIFAR100 and TinyImageNet instances, $T$ is set to 250 seconds. We crafted 100

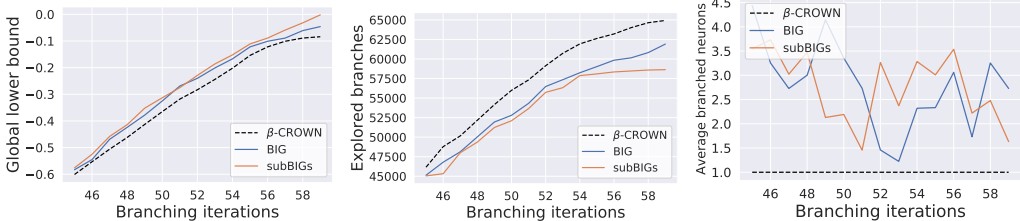

Figure 5: {**(Left)** Global lower bounds; **(Middle)** Total explored branches; **(Right)** Average used bound implications} trends along with branching iterations on one specific instance in **VeriHard**-MNIST-A-Adv benchmark. More than one neurons are branched per iteration due to BIG, and the bounds and the number explored branches both improve noticeably.

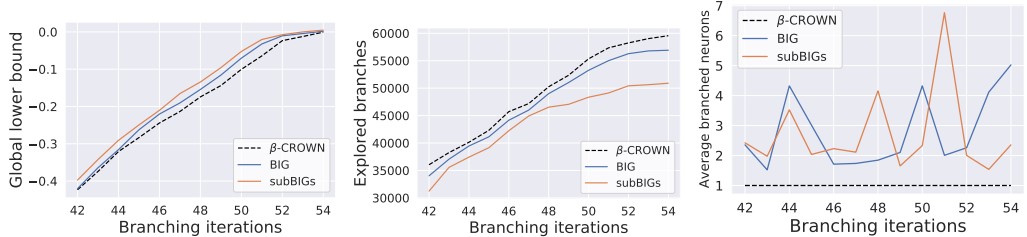

Figure 6: {**(Left)** Global lower bounds; **(Middle)** Total explored branches; **(Right)** Average used bound implications} trends along with branching iterations on one specific instance in **VeriHard**-CIFAR10-A-Adv benchmark.

instances from each benchmark, assigning the respective $\epsilon$ to each instance following the algorithm mentioned before. We will release our benchmark in the standard VNNLIB format.

### E.2 EXPERIMENTAL SETUP

Our implementation is based on the open-source $\alpha, \beta$-CROWN verifier by integrating BIG and subBIGs construction and branching utilization code into that. For BIG and subBIGs construction, we select top-$K$ implicant neurons to derive our paired bound implications as mentioned in Theorem 3.2 with $K = 1000$. Specifically for subBIGs, we use first $M = 8$ splits by BaB and apply a 50-steps gradient descent with Adam optimizer to optimize our objective $l^*_{\text{relaxed-multi}}$ on up to $2^M = 256$ unsolved subproblems with initial learning rate as $0.1$ and its decay factor as $0.99$. Throughout our experiments, we employ Filtered Smart Branching (FSB) (De Palma et al., 2021a) as our branching heuristics and apply Adam optimizer to optimize both $\alpha$ and $\beta$ for 20 iterations during verification process. The initial learning rate is set to be $0.1$ for $\alpha$ optimization and $0.05$ for $\beta$, while the corresponding decay ratio is set to be $0.995$ for $\alpha$ and $0.98$ for $\beta$. All our experiments are conducted on one NVIDIA A100 GPU device (80G memory). Timeout for classic benchmark is aligned with prior work: MNIST-CNN-A-Adv (200s), CIFAR10-CNN-A-Adv (200s), CIFAR10-CNN-A-Mix (200s), MNIST-ConvSmall (180s), CIFAR10-ConvSmall (180s). For **VeriHard** benchmark, we set timeout to be 200 seconds for MNIST and CIFAR10 instances and 250 second for CIFAR100 and TinyImageNet instances.

## F VISUALIZATION OF BIG-ENHANCED BAB PROCESS

In this section, we provide more figures showing the bounds improvements, the number of branches, and the average number of branched neurons per BaB iteration, on different benchmarks and data-points.

## G ABLATION STUDIES ON GCP-CROWN

GCP-CROWN (Zhang et al., 2022) enhances NN verification efficiency by introducing general cutting plane methods through mixed integer programming (MIP) solver. The derived cutting

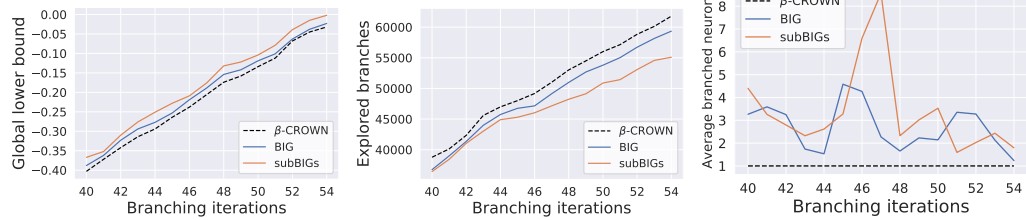

Figure 7: {(**Left**) Global lower bounds; (**Middle**) Total explored branches; (**Right**) Average used bound implications} trends along with branching iterations on one specific instance in **VeriHard**-CIFAR10-A-Mix benchmark.

plane constraints could also help reduce the number of branches within verification. We conduct experiments on showing the average number of branches on various benchmarks by utilizing $\beta$-CROWN, GCP-CROWN, BIG, and the combined approach GCP-CROWN+BIG. Results are shown in Table 5. It is worth noting that GCP-CROWN heavily relies on a MIP solver to find cutting planes, and for CIFAR100 and TinyImageNet models, GCP-CROWN cannot provide useful cutting planes due to its high computational cost.

Table 5: Total number of branches (less is better) on MNIST, CIFAR10, CIFAR100, and TinyImageNet benchmarks. GCP-CROWN+BIG could further reduce the number of branches compared with GCP-CROWN only. Note that GCP-CROWN cannot scale to CIFAR100 and TinyImageNet, yet our bound implications can still bring improvements.

| Number of Branches | $\beta$-CROWN | GCP-CROWN | BIG | GCP-CROWN+BIG |
|---|---|---|---|---|
| MNIST-CNN-A-Adv | 53120.9 | 22250.4 | 45130.4 | **21230.4** |
| CIFAR10-CNN-A-Mix | 50614.2 | 24103.7 | 46132.4 | **23792.5** |
| CIFAR10-CNN-A-Adv | 55764.8 | 28607.1 | 50014.2 | **28204.6** |
| CIFAR100-small | 74113.5 | - | 70006.6 | - |
| TinyImageNet-medium | 64253.9 | - | 60082.5 | - |

From this table we can see that, GCP-CROWN+BIG could further reduces branch counts compared to GCP-CROWN alone. Note that GCP-CROWN cannot scale to CIFAR100 and TinyImageNet, yet our bound implications can still bring improvements.

We also exported the cutting planes from GCP-CROWN and compared them to our implications on the same set of instances. We report the average number of implications found by us, the average number of cutting planes in GCP-CROWN, and the overlaps. Here overlap means that GCP-CROWN finds a constraint mentioning the same pair of two neurons as we found as implicated and implicant neurons. Results are shown in Table 6.

Table 6: The average number of implications in BIG, GCP-CROWN, and the average number of overlaps with the cutting plane constraints in GCP-CROWN. Here overlap means that GCP-CROWN finds a constraint mentioning the same pair of two neurons as we found as implicated and implicant neurons.

| Average Implications | GCP-CROWN | BIG | Overlaps |
|---|---|---|---|
| MNIST-CNN-A-Adv | 574.2 | 761.7 | 102.2 |
| CIFAR10-CNN-A-Mix | 683.7 | 1103.3 | 89.9 |
| CIFAR10-CNN-A-Adv | 796.9 | 1289.7 | 104.2 |

As we can see, the implications only take a minority of the overall implications we found with our method. This is expected because GCP-CROWN uses a MIP solver to generate cutting planes, which is unaware of the underlying neural network architecture and uses generic methods (such as Gomory cuts) to find implications among variables.

## H   LIMITATIONS

Our method suffers from several limitations which are commonly existed by most existing BaB-based verification methods (Wang et al., 2021; Ferrari et al., 2021; Zhang et al., 2022; Xu et al., 2020b), and is limited by the capability of existing verifiers, which focuses on branch-and-bound

for ReLU networks only. For other non-linear activations and non-perception networks (such as Transformers), applying our method to the very recent work on BaB-based verification for general non-linear functions (Shi et al., 2023) could be an interesting future work. Our method also cannot support the verification of very large neuron networks, such as NNs with billions of parameters. Improving the overall computational efficiency could be our future direction.

---

**Algorithm 3** Construct BIG

---

1: **Inputs**: constraints of all unstable neurons $\underline{\mathbf{a}}, \overline{\mathbf{a}} \in \mathbb{R}^{N \times d_0}$ and $\underline{c}, \overline{c} \in \mathbb{R}^N$, where $N$ is number of unstable neurons and $d_0$ is the length of the model input $\boldsymbol{x}_0$; perturbation size $\epsilon, K$
2: **Outputs**: BIG: $\mathbf{G}(\mathbf{V}, \mathbf{E})$
3: $\underline{\mathbf{S}}, \overline{\mathbf{S}} \leftarrow \text{TOPKFILTERING}(\underline{\mathbf{a}}, \overline{\mathbf{a}}, \underline{c}, \overline{c}, \boldsymbol{x}_0, \epsilon, K)$    ▷ Get indices of Top-$K$ constraints from Alg. 2
4: **for** $p = 1$ to $N$ **do**                                             ▷ Iterate implicant neurons
5:     **for** $q = 1$ to $N$ **do**                                     ▷ Iterate implicated neurons
6:         **if** $p \in \underline{\mathbf{S}}$ **then**
7:             ▷ Consider inactive implicant constraint to improve lower bound
8:

$$x_k^* := \begin{cases} x_{0,k} - \epsilon \cdot \underline{a}_k^{(i_q, j_q)}, & \text{if } \underline{a}_k^{(i_q, j_q)} > 0 \\ x_{0,k} + \epsilon \cdot \underline{a}_k^{(i_q, j_q)}, & \text{if } \underline{a}_k^{(i_q, j_q)} < 0 \\ x_{0,k} - \epsilon \cdot \text{sign}(\underline{a}_k^{(i_p, j_p)}), & \text{if } \underline{a}_k^{(i_q, j_q)} = 0 \end{cases}$$

9:             **if** $\underline{\mathbf{a}}^{(i_p, j_p)^\top} x^* + \underline{c}^{(i_p, j_p)} > 0$ **then**          ▷ Useful implicant constraint
10:                 $l_{\text{relaxed}}^* \leftarrow$ Solve Eq. 6 with the implicated neurons: $\underline{\mathbf{a}}^{(i_q, j_q)^\top} x + \underline{c}^{(i_q, j_q)}$
11:                 **if** $l_{\text{relaxed}}^* \geq 0$ **then**                    ▷ $z_{j_q}^{(i_q)}$ become to active
12:                     $\mathbf{E} \leftarrow \mathbf{E} \cup (Q_{j_p}^{(i_p)}, P_{j_q}^{(i_q)})$       ▷ add edge $(Q_{j_p}^{(i_p)}, P_{j_q}^{(i_q)})$ to the graph

13:             ▷ Consider inactive implicant constraint to improve upper bound
14:

$$x_k^* := \begin{cases} x_{0,k} - \epsilon \cdot \overline{a}_k^{(i_q, j_q)}, & \text{if } \overline{a}_k^{(i_q, j_q)} > 0 \\ x_{0,k} + \epsilon \cdot \overline{a}_k^{(i_q, j_q)}, & \text{if } \overline{a}_k^{(i_q, j_q)} < 0 \\ x_{0,k} - \epsilon \cdot \text{sign}(\underline{a}_k^{(i_p, j_p)}), & \text{if } \underline{a}_k^{(i_q, j_q)} = 0 \end{cases}$$

15:             **if** $\underline{\mathbf{a}}^{(i_p, j_p)^\top} x^* + \underline{c}^{(i_p, j_p)} > 0$ **then**          ▷ Useful implicant constraint
16:                 $l_{\text{relaxed}}^* \leftarrow$ Solve Eq. 10 with the implicated neurons: $\overline{\mathbf{a}}^{(i_q, j_q)^\top} x + \overline{c}^{(i_q, j_q)}$
17:                 **if** $l_{\text{relaxed}}^* \leq 0$ **then**                    ▷ $z_{j_q}^{(i_q)}$ become to inactive
18:                     $\mathbf{E} \leftarrow \mathbf{E} \cup (Q_{j_p}^{(i_p)}, Q_{j_q}^{(i_q)})$       ▷ add edge $(Q_{j_p}^{(i_p)}, Q_{j_q}^{(i_q)})$ to the graph

19:         **if** $p \in \overline{\mathbf{S}}$ **then**
20:             ▷ Consider active implicant constraint to improve lower bound
21:

$$x_k^* := \begin{cases} x_{0,k} - \epsilon \cdot \underline{a}_k^{(i_q, j_q)}, & \text{if } \underline{a}_k^{(i_q, j_q)} > 0 \\ x_{0,k} + \epsilon \cdot \underline{a}_k^{(i_q, j_q)}, & \text{if } \underline{a}_k^{(i_q, j_q)} < 0 \\ x_{0,k} + \epsilon \cdot \text{sign}(\overline{a}_k^{(i_p, j_p)}), & \text{if } \underline{a}_k^{(i_q, j_q)} = 0 \end{cases}$$

22:             **if** $\overline{\mathbf{a}}^{(i_p, j_p)^\top} x^* + \overline{c}^{(i_p, j_p)} < 0$ **then**          ▷ Useful implicant constraint
23:                 $l_{\text{relaxed}}^* \leftarrow$ Solve Eq. 11 with the implicated neurons: $\underline{\mathbf{a}}^{(i_q, j_q)^\top} x + \underline{c}^{(i_q, j_q)}$
24:                 **if** $l_{\text{relaxed}}^* \geq 0$ **then**                    ▷ $z_{j_q}^{(i_q)}$ become to inactive
25:                     $\mathbf{E} \leftarrow \mathbf{E} \cup (P_{j_p}^{(i_p)}, Q_{j_q}^{(i_q)})$       ▷ add edge $(P_{j_p}^{(i_p)}, Q_{j_q}^{(i_q)})$ to the graph

26:             ▷ Consider active implicant constraint to improve upper bound
27:

$$x_k^* := \begin{cases} x_{0,k} - \epsilon \cdot \overline{a}_k^{(i_q, j_q)}, & \text{if } \overline{a}_k^{(i_q, j_q)} > 0 \\ x_{0,k} + \epsilon \cdot \overline{a}_k^{(i_q, j_q)}, & \text{if } \overline{a}_k^{(i_q, j_q)} < 0 \\ x_{0,k} + \epsilon \cdot \text{sign}(\overline{a}_k^{(i_p, j_p)}), & \text{if } \overline{a}_k^{(i_q, j_q)} = 0 \end{cases}$$

28:             **if** $\overline{\mathbf{a}}^{(i_p, j_p)^\top} x^* + \overline{c}^{(i_p, j_p)} < 0$ **then**          ▷ Useful implicant constraint
29:                 $l_{\text{relaxed}}^* \leftarrow$ Solve Eq. 12 with the implicated neurons: $\overline{\mathbf{a}}^{(i_q, j_q)^\top} x + \overline{c}^{(i_q, j_q)}$
30:                 **if** $l_{\text{relaxed}}^* \leq 0$ **then**                    ▷ $z_{j_q}^{(i_q)}$ become to active
31:                     $\mathbf{E} \leftarrow \mathbf{E} \cup (P_{j_p}^{(i_p)}, P_{j_q}^{(i_q)})$       ▷ add edge $(P_{j_p}^{(i_p)}, P_{j_q}^{(i_q)})$ to the graph

32: **Return:** $\mathbf{G}(\mathbf{V}, \mathbf{E})$

---

---

**Algorithm 4** Generate subproblems in one BaB iteration with BIG.

---

1: **Inputs**: $\mathbf{G}(\mathbf{V}, \mathbf{E})$, selected unstable neuron $\boldsymbol{z}_j^{(i)}$

2: **Outputs**: BIG-generated subproblems.

3: $\mathbf{S}_P \leftarrow \{P_j^{(i)}\}, \mathbf{S}_Q \leftarrow \{Q_j^{(i)}\}$             ▷ traverse BIG starting from node $P_j^{(i)}$ and $Q_j^{(i)}$

4: **while** $\mathbf{S}_P$ has unvisited nodes **do**

5:      Pick a unvisited node $X_{j'}^{(i')}$ out from $\mathbf{S}_p$

6:      **for** $e = (X_{j'}^{(i')}, Y_{j*}^{(i^*)}) \in \mathbf{E}$ **do**

7:          $\mathbf{S}_P \leftarrow \mathbf{S}_P \cup \{Y_{j*}^{(i^*)}\}$

8:      Mark node $X_{j'}^{(i')}$ as visited

9: **while** $\mathbf{S}_Q$ has unvisited nodes **do**

10:      Pick a unvisited node $X_{j'}^{(i')}$ out from $\mathbf{S}_Q$

11:      **for** $e = (X_{j'}^{(i')}, Y_{j*}^{(i^*)}) \in \mathbf{E}$ **do**

12:          $\mathbf{S}_Q \leftarrow \mathbf{S}_Q \cup \{Y_{j*}^{(i^*)}\}$

13:      Mark node $X_{j'}^{(i')}$ as visited

14: Add splits $z_{j'}^{(i')} \geq 0$ (or $z_{j'}^{(i')} \leq 0$) from nodes in $\mathbf{S}_P$ to subproblem $z_j^{(i)} \geq 0$

15: Add splits $z_{j'}^{(i')} \leq 0$ (or $z_{j'}^{(i')} \leq 0$) from nodes in $\mathbf{S}_Q$ to subproblem $z_j^{(i)} \leq 0$

---

