5: $\underline{\mathbf{a}}_{\text{sorted}}^{(i_2,j_2)} \leftarrow \{\underline{\mathbf{a}}_{\boldsymbol{I}_1}^{(i_2,j_2)} \cdot \epsilon, \underline{\mathbf{a}}_{\boldsymbol{I}_2}^{(i_2,j_2)} \cdot \epsilon, ...., \underline{\mathbf{a}}_{\boldsymbol{I}_{d_0}}^{(i_2,j_2)} \cdot \epsilon\}$    ▷ Sort $\underline{\mathbf{a}}^{(i_2,j_2)}$ by index $I$ and scale by $\epsilon$

6: $\left(\underline{\mathbf{a}}_{-}^{(i_2,j_2)}\right)_i \leftarrow -\sum_{k=1}^{i}(|\underline{\mathbf{a}}_{\text{sorted}}^{(i_2,j_2)}|_k), \; i \in [d_0]$    ▷ Cumulative sum of $|\underline{\mathbf{a}}_{\text{sorted}}^{(i_2,j_2)}|$

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

---