# OpenReview forum: "Improving Branching in Neural Network Verification with Bound Implication Graph"
_ICLR.cc/2024/Conference — Submitted to ICLR 2024_

### Official Review · Reviewer_Wsyq · 2023-10-27

**Soundness:** 3 good
**Presentation:** 2 fair
**Contribution:** 3 good
**Rating:** 6
**Confidence:** 4

**Summary:**

This paper proposes a procedure for eliminating infeasible sub-problems in branch-and-bound-based neural network verification procedures.

**Strengths:**

- The paper considers an important problem, verifying neural networks.
- The paper introduces a new procedure that heuristically select a subset of neurons and keep track of their dependencies during the search, which can be used to remove sub-problems and tighten bounds.
- The paper implemented the idea in $\alpha$-$\beta$-CROWN and obtained consistent performance gain on a set of robustness verification tasks.
- The paper is well-written.

**Weaknesses:**

- The paper only considers certifying adversarial robustness on perception networks.
- The general idea of removing sub-problems during case-analysis-based search by recording the dependencies between case splits has been considered in the past in the neural network verification setting, such as Planet. Therefore, the technical contribution of this paper is rather incremental and specific to $\alpha$-$\beta$-CROWN.


[1] Ehlers, Ruediger. "Formal verification of piece-wise linear feed-forward neural networks." Automated Technology for Verification and Analysis: 15th International Symposium, ATVA 2017, Pune, India, October 3–6, 2017, Proceedings 15. Springer International Publishing, 2017.

**Questions:**

- What is the actual perturbation bounds and perturbation dimensions on the VeriHard benchmark sets?
- Are the techniques described in the paper applicable beyond $\alpha$-$\beta$-CROWN, such as in NNV and Marabou?

---

> ### Author Response · Authors · 2023-11-22
> **Thank you for your encouraging comments and we answer your questions below. (Part 1/2)**
>
> We greatly appreciate the reviewer for recognizing the strengths of our work. We answer your questions about the limitations of our work, details about the new benchmark, and applicability to other verifiers. We also provide **new experiments** comparing our method to prior work which also used case-analysis-based approaches to find neuron implications. We hope the reviewer can reevaluate our paper based on the new evidence provided.
>
> > The paper only considers certifying adversarial robustness on perception networks.
>
> Our method largely relied on state-of-the-art BaB-based verification frameworks such as [2,3,4,5,6], and is limited by the capability of existing verifiers, which focuses on branch-and-bound (BaB) for ReLU networks only. We want to point out that this is a limitation of the majority of verification papers. For other non-linear activations and non-perception networks (such as Transformers), one very recent workshop paper [1] (still unpublished) considered BaB. Our method can be extended to the generalized BaB in [1], by changing the right-hand-side constraint in Eq. (5) and (6) to a non-zero splitting point. We will work towards this direction in our future work.
>
> > The general idea of removing sub-problems during case-analysis-based search has been considered in Planet.
>
> We acknowledge that prior work such as Planet (2017) considered case analysis in the same spirit. We revised our paper and discussed this reference in the relevant part in Section 5. We want to emphasize that our main contribution is a new optimization-based formulation that is more powerful and less restrictive to find bound implications. In our revised paper, **we added a more recent baseline**, Venus2 (2021) [7,8], which also finds dependencies among neurons but with a much weaker and simpler method compared to ours.
>
> We highlighted the new results below (more details added to Section 4). Following [7,8], here we divide the implications into inter-layer (implicated and implicant from different layers) and intra-layer (from the same layer) ones. **We outperform the baselines by finding 7x - 180x more implications**. So although some existing work also considered similar analysis, ours is significantly stronger.
>
> |   | MNIST-MLP | CIFAR-MLP | CIFAR-CNN-A-Adv | CIFAR-CNN-A-Mix |
> |----------------------|:---------:|:---------:|:---------------:|:---------------:|
> | Venus2 (inter-layer) |   56.15   |    4.66   |       6.42      |      19.17      |
> | Ours (inter-layer)   |   203.5   |   397.5   |      340.0      |      486.5      |
> | Venus2 (intra-layer) |    9.80   |   35.33   |        0        |        0        |
> | Ours (intra-layer)   |   263.4   |   502.1   |      763.4      |      803.2      |
>
> > The technical contribution of this paper is rather incremental
>
> Besides the **strong empirical improvements** (7x - 170x improvement) shown above, our paper also makes a few unique technical contributions:
>
> 1. Our first theoretical contribution is the formulation of finding implications constraints as a generic optimization problem. The optimization formulation did not exist in prior work, and is a stronger formulation than previous work, enabling finding implications between any two neurons (prior work [7,8] was restricted to neurons in the same layer or consecutive layers).
>
> 2. Our second theoretical contribution is a fast way of solving these optimization problems, by reusing bound equations that were readily available during the solving process without recomputation. None of the existing work shows that the bound equations (for free) can be used to find neuron implications and improve verification.
>
> 3. Our method also works significantly better on convolutional networks due to its technical advantage. The technique in [7,8] considers implications using weak intermediate layer bounds of one convolutional layer only, and very few implications can be found because the convolutional layer weights are sparse and uncorrelated (**zero** reported in Table 1) . In contrast, our optimization-based formulation found **hundreds of implications**.
>
> 4. In addition, we also demonstrated multiple subproblem-specific bound implication graphs can be built to further enhance its effectiveness.

---

> ### Author Response · Authors · 2023-11-22
> **Thank you for your encouraging comments and we answer your questions below. (Part 2/2)**
>
> > The technical contribution of this paper is specific to CROWN. Are the techniques described in the paper applicable beyond ab-CROWN, such as in NNV and Marabou?
>
> Our algorithm to find the bound implication is independent of ab-CROWN, as it only takes linear bound equations and then returns the implication graph. The graph can potentially be used in many BaB-based verifiers, including **Marabou, Mn-BaB, VeriNet, and Oval**. Some of these verifiers also have similar linear bounds so we can obtain them without additional cost, and for the verifiers that do not use these linear bounds, a single pass of CROWN bound propagation (very fast) is sufficient to get these bounds to enable the benefits of bound implications in these verifiers. The only exception is NNV, where it is unclear to us whether BaB over ReLU neurons was used. We will provide our implementation in a separate module, independent of ab-CROWN so it may potentially be used in other verifiers.
>
> > What is the actual perturbation bounds and perturbation dimensions on the VeriHard benchmark sets?
>
> Thanks for pointing this out. We include the perturbation bounds and dimensions on the VeriHard benchmark sets in an [anonymous Google sheet](https://docs.google.com/spreadsheets/d/1hHL9_03N7zEAAyLRwfasTBwYXHEIRG-pBetcYtS7hqE/edit?usp=sharing) so you can take a look. We will publish our dataset once our paper is accepted.
>
> References:
>
> [1] Shi, Zhouxing, et al. "Formal Verification for Neural Networks with General Nonlinearities via Branch-and-Bound." 2nd Workshop on Formal Verification of Machine Learning (WFVML 2023). 2023.
>
> [2] Katz, Guy, et al. "The marabou framework for verification and analysis of deep neural networks." Computer Aided Verification: 31st International Conference, CAV 2019, New York City, NY, USA, July 15-18, 2019, Proceedings, Part I 31. Springer International Publishing, 2019.
>
> [3] Wang, Shiqi, et al. "Beta-crown: Efficient bound propagation with per-neuron split constraints for neural network robustness verification." Advances in Neural Information Processing Systems 34 (2021): 29909-29921.
>
> [4] Ferrari, Claudio, et al. "Complete verification via multi-neuron relaxation guided branch-and-bound." arXiv preprint arXiv:2205.00263 (2022).
>
> [5] Zhang, Huan, et al. "General cutting planes for bound-propagation-based neural network verification." Advances in Neural Information Processing Systems 35 (2022): 1656-1670.
>
> [6] Xu, Kaidi, et al. "Fast and complete: Enabling complete neural network verification with rapid and massively parallel incomplete verifiers." arXiv preprint arXiv:2011.13824 (2020).
>
> [7] Kouvaros, Panagiotis, and Alessio Lomuscio. "Towards Scalable Complete Verification of Relu Neural Networks via Dependency-based Branching." IJCAI. 2021.
>
>
> [8] Botoeva, Elena, et al. "Efficient verification of relu-based neural networks via dependency analysis." Proceedings of the AAAI Conference on Artificial Intelligence. Vol. 34. No. 04. 2020.

---

> > ### Comment · Reviewer_Wsyq · 2023-11-22
> >
> > I thank the authors for the clarifications of their contributions and additional information. I would like to keep my score.

---

### Official Review · Reviewer_HQLe · 2023-10-30

**Soundness:** 3 good
**Presentation:** 1 poor
**Contribution:** 2 fair
**Rating:** 5
**Confidence:** 5

**Summary:**

The paper incorporates implication constraints for the states of the ReLUs into
branching mechanisms for branch-and-bound-based neural network verification. An
implication constraint between two ReLUs expresses that if one them is stable
in a state then the other one has to be stable in a state as well. The
constraints are computed from the  bound equations of the ReLUs  and are then
exploited in branch-and-bound frameworks to reduce the total number of
branches that need to be explored.

**Strengths:**

The paper adapts notions from previously developed notions [1, 2] to
branch-and-bound frameworks and shows that they help improve verification
times.

**Weaknesses:**

1. The paper highly incremental to [1, 2].

2. It often presents several ideas as novel when they were first discussed in
[1, 2]. The following comments illustrate this weakness.

    a. While the experimental results from [1] clearly  show that implication
    constraints help with verification times,  the paper says "they were
    handled as constraints to a black-box MIP solver without validating their
    usefulness".

    b. Bound implication graphs were previously constructed in [2], where they
    were precisely used to reduce the total number of branches. The paper
    however presents the implication graph as a novel aspect of the paper and
    does not reference [2].

    c. While the procedures computing the implication constraints in [1] are
    GPU-friendly the paper claims the opposite. Although they were used as MILP
    constraints in [1] and as a branching heuristic in [2], their
    identification relies only on matrix operations.  [2] discusses that they
    can be incorporated into any verifier with  a branching mechanism.

    d. The paper does not clarify with the introduction of the implication
    constraints that they were first introduced in [1].

3. The main novelty of the paper is the computation of the implication
constraints using bound equations (which are computed using previously
established methods) instead of the concrete bounds. The delta is small in my
opinion for an ICLR paper.

4. The implication constraint identification procedures are not compared with
the ones from [1]. It is not clear that they perform better. For instance [1,
2] show similar gains. Also, the procedures from [1] should be faster because
they deal with concrete bounds instead of bound equations.

[1] E. Botoeva, et al. Efficient Verification of ReLU-based Neural Networks via
Dependency Analysis. AAAI 2020.
[2] P. Kouvaros, A. Lomuscio. Towards Scalable Complete Verification of ReLU
Neural Networks via Dependency-based Branching. IJCAI 21.

**Questions:**

See comments above.

---

> ### Author Response · Authors · 2023-11-22
> **We revised our entire paper to reflect your mentioned papers, fixed inaccurate claims, and added new experiments.  (Part 1/3)**
>
> We thank the reviewer for giving detailed technical explanation for your mentioned work [1,2]. We have significantly revised our paper based on your suggestions, and correctly **cited and discussed [1,2] throughout the paper**; major changes were marked in blue.  We now discuss [1,2] early in our paper in the introduction, do not claim the implication graph as part of our contribution in Section 3, and remove imprecise sentences in Section 5. We also added a **detailed experimental comparison to [1,2]** to show our strong improvements.
>
> The main novelty of our paper is not about constructing the graph, but a new formulation that is more powerful and less restrictive to find bound implications. **Compared to [1,2], we have both theoretical advantage and strong experimental evidence with 7x - 170x more bound implications found** (Table 1 in our revised paper). We respectfully disagree this is a “small delta” or incremental contribution. We hope the reviewer can reevaluate our paper based on new discussions and experiments about [1,2], where we have made it clear that [1,2] are important prior work in this field.
>
> > The main novelty of the paper is the computation of the implication constraints using bound equations (which are computed using previously established methods) instead of concrete bounds.
>
> We thank the reviewer for recognizing that part of our contribution was to use bound equations to solve bound implications. However, this is only a small part, and our method includes **multiple novel contributions that were never discussed or demonstrated**:
>
> 1. Our first theoretical contribution is the formulation of finding implications constraints as a generic optimization problem. The optimization formulation did not exist in prior work, and is a stronger formulation than previous work, enabling finding implications between any two neurons. We discuss the theoretical advantage of our work in Section 5 and also in this response (a few paragraphs below).
>
> 2. Our second theoretical contribution is a fast way of solving these optimization problems, by reusing bound equations that were readily available during the solving process without recomputation. None of the existing work shows that the bound equations can be used to find neuron implications. In addition, as you mentioned, these bound equations are “previously established” and have been widely used in many verifiers, and here our novel usage of these equations in a new setting shows potential wide applications of our approach in many verifiers.
>
> 3. Empirically, we demonstrated that our new method of finding implications is significantly better than [1,2]. In our added experiments, we used the same networks as evaluated in [1,2], and we reported 7x - 170x more inter-layer and intra-layer implications (see details below).
>
> 4. Our method also works **significantly better on convolutional networks** due to its technical advantage. The technique in [1,2] considers implications using intermediate layer bounds of one convolutional layer only, and very few implications can be found because the convolutional layer weights are sparse and uncorrelated (**zero** reported in Table 1). In contrast, our optimization-based formulation found **hundreds of implications**.

---

> ### Author Response · Authors · 2023-11-22
> **We revised our entire paper to reflect your mentioned papers, fixed inaccurate claims, and added new experiments.  (Part 2/3)**
>
> > The implication constraint identification procedures are not compared with the ones from [1]. It is not clear that they perform better.
>
> We added comparisons in both experiments (Section 4) and in theoretical perspectives (Section 5).
>
> **Experimental comparison to [1,2]**: We added experiments in Section 4 to compare to [1, 2] and showed that we could find significantly more neuron implications. We evaluate our method on a few networks used in [1,2], and we use the official code of Venus2. To allow a fair comparison, here we do not use any particular BaB solver and consider the implications found in the initial verification problem. The results are presented in Table 1 in our revised paper and also highlighted below. We outperform the baselines by finding 7x - 180x more implications. The difference is especially significant in convolutional networks - the zero results from Venus2 are not errors (see our contribution 4 above).
>
> |   | MNIST-MLP | CIFAR-MLP | CIFAR-CNN-A-Adv | CIFAR-CNN-A-Mix |
> |----------------------|:---------:|:---------:|:---------------:|:---------------:|
> | Venus2 (inter-layer) |   56.15   |    4.66   |       6.42      |      19.17      |
> | Ours (inter-layer)   |   203.5   |   397.5   |      340.0      |      486.5      |
> | Venus2 (intra-layer) |    9.80   |   35.33   |        0        |        0        |
> | Ours (intra-layer)   |   263.4   |   502.1   |      763.4      |      803.2      |
>
> **Theoretical comparison to [1,2]**: Compared to [1,2], our formulation is significantly stronger, for the following reasons:
>
> 1. For inter-layer dependency, [1,2] can only consider the case where a neuron in one layer is set to be inactive (0), and propagation the bound to the layers following it. In Lemma 3 of [2], cases 3 and 4 are empty, indicating that they cannot find any implications when one neuron is set to be active. Our methods can be used to find all four kinds of implications.
>
> 2. Furthermore, our method can find implicated neurons *before* the implicant neuron. For example, by setting a neuron in layer 3 to be active or inactive, we may imply that a neuron in layer 1 or 2 (before layer 3) is active or inactive. This is impossible in [1, 2].
>
> 3. For intra-layer dependency, our methods use bound equations that are propagated to the input layer, which utilizes tighter bounds on the input layer rather than the relatively looser intermediate layer bounds in [1,2]. Our formulation is also optimization-based, allowing us to find the strongest implications under this formulation. As a result, we found significantly more intra-layer neuron dependencies (Table 1).
>
> 4. Furthermore, the method in [1,2] does not work well for convolutional networks, because the equivalent weight matrices are sparse, so the pair of neurons under investigation may have no correlation at all  (Table 1 where Venus2 reports 0). Our method uses the bound equations at the input x, which is often dense (as large as the receptive field of the entire network, rather than one layer) and can capture neuron correlations better.

---

> ### Author Response · Authors · 2023-11-22
> **We revised our entire paper to reflect your mentioned papers, fixed inaccurate claims, and added new experiments. (Part 3/3)**
>
> We have also addressed each individual weakness you pointed out:
>
> > a. While the experimental results from [1] clearly show that implication constraints help with verification times, the paper says "they were handled as constraints to a black-box MIP solver without validating their usefulness".
>
> We removed this confusing sentence. Our new characterization of [1,2] is provided throughout the paper, particularly in Sections 1 and 5.
>
> > b. Bound implication graphs were previously constructed in [2], where they were precisely used to reduce the total number of branches. The paper however presents the implication graph as a novel aspect of the paper and does not reference [2].
>
> We removed this claim throughout the paper, cited [2], and rewrote the relevant paragraphs in section 3. Our new formulation to find neuron implications is an important theoretical contribution, and we outperformed [2] by finding 7x - 170x more implications. This is not an incremental contribution.
>
> > c. While the procedures computing the implication constraints in [1] are GPU-friendly the paper claims the opposite.
>
> Although [1,2] did not use a GPU implementation, we acknowledge that [1,2] can potentially be applied to any verifiers and we removed relevant claims in our paper.
>
> > d. The paper does not clarify with the introduction of the implication constraints that they were first introduced in [1].
>
> We rewrote the introduction and cited [1,2] at the beginning.
>
> We thank the reviewer again for discussing the details in relevant papers [1,2], and we hope all your concerns are now addressed. We hope the new representation of our paper has made it clear that [1,2] is important prior work in this field. Please let us know if you have any further concerns, thank you.
>
> References:
>
> [1] E. Botoeva, et al. Efficient Verification of ReLU-based Neural Networks via Dependency Analysis. AAAI 2020.
>
> [2] P. Kouvaros, A. Lomuscio. Towards Scalable Complete Verification of ReLU Neural Networks via Dependency-based Branching. IJCAI 21.

---

### Official Review · Reviewer_2ULS · 2023-10-30

**Soundness:** 3 good
**Presentation:** 2 fair
**Contribution:** 3 good
**Rating:** 5
**Confidence:** 4

**Summary:**

This work proposes to leverage "bound implications" during neural network verification to reduce the number of explored branches in a BaB procedure. To this end, they construct a bound implication graph where each node corresponds to a positive or negative split of a formerly unstable neuron/ReLU and edges correspond to a split of the outgoing node implying a stabilization of the incoming node. They find these implications by evaluating the linear bounds on the implicated neuron, as computed during LiRPA, over the part of the specification compatible with a linear relaxation of the split of the implicating neuron. Enforcing these implied splits during BaB, improves verification performance on a wide range of tasks.

**Strengths:**

* The tackled issue of (certified) adversarial robustness is of high importance.
* To the best of my knowledge, this paper is the first to propose bound implications between arbitrary layers and among more than two neurons.
* Combining (sub)BIG with established BaB-based verification methods consistently improves their performance, in some cases significantly so.
* The novel VeriHard benchmark is not only useful for this work but also to benchmark further novel certification methods.
* The two-stage check for the potential of bound implication (Theorem 3.2) is elegant and helps to significantly reduce the computational cost for finding bound implications.

**Weaknesses:**

* An important baseline, GCP (Zhang et al. 2022), the most recent version of $\alpha\beta$-CROWN, is missing from all comparisons. However the MILP constraints leveraged by this approach might be highly correlated to the bound implications described in this work. Thus, a direct comparison or even better combination with GCP would be essential to judge the marginal contribution of this work over state-of-the-art methods.
* Clarity of presentation including copy writing (see some examples below) could be improved significantly.
* Significance of improvements on established benchmarks (less than $1\%$ is unclear).

**Questions:**

### Questions
1) Have you experimented with considering the bound implication as part of the split heuristic? E.g. adding the scores of implicated neurons to the implicating one.
2) Did you investigate the interaction of these bound implications with the general cutting planes from GCP (also based on beta-CROWN)? It seems like similar implications might be captured by the MILP solver used there.
3) Did you investigate the effect of how many implicating neurons $K$ you consider?
4) In Theorem 3.2, it seems like the two enumerated points add an additional (exhaustive) case distinction if Equation (7) holds, where the first one can be checked independently of the implicated neuron. I would consider moving them to a separate Corollary. What exactly does it mean for its “linear bounds to not have an intersection with C”? Is the first constraint (functionally) equivalent to $a x_0 + c + \epsilon ||a|| < 0 $, i.e. the lower bound will remain negative for any $x \in \mathcal{C}$? I believe the presentation of Theorem 3.2 could be improved significantly. In particular, it is not initially clear, that either of the two cases follows from Equation (7) rather than being additional constraints.

### Comments
* The scaling of the y-axis in Figure 2 is unclear making it hard to interpret. I would consider a linear scale instead.
* I believe it should be "(inputs $\leq$ 0)" for the "inactive" case in Section 1 paragraph 3.
* An illustration of Theorem 3.2 could help communicate the underlying intuition.

#### Typos
- Mn-BaB instead of MN-BaB in Section 1 paragraph 2
- Extra “the” in the third line of Section 2
- The second sentence after Equation (3) is broken and missing a full-stop.
- “Activate and inactive” third last line in Section 2

### Conclusion
The proposed idea of using a (sub)BIG to reduce the number of to be considered branches in BaB-based neural network verification is novel and seems effective. Further, the novel VeriHard benchmark might prove valuable for evaluating novel certification methods beyond this work. However, the lacking comparison to the important baseline GCP makes both its complementarity with and performance compared to current state-of-the-art methods unclear, preventing me from recommending (strong) acceptance

---

> ### Author Response · Authors · 2023-11-22
> **We added GCP-CROWN and branching heuristic results as you suggested and improved clarity (Part 1/2)**
>
> Thanks for your valuable comments and constructive feedback. We added comparisons to GCP-CROWN as you suggested, and also improved the clarity of theorems. We carefully addressed your comments below, and we hope the reviewer can reevaluate our paper based on these new results.
>
> > a direct comparison or even better combination with GCP would be essential
>
> Thanks for pointing this out. We carefully reviewed the GCP-CROWN paper you mentioned and we also integrated BIG into GCP-CROWN. However, we want to emphasize that GCP-CROWN heavily relies on a MIP solver to find cutting planes, which cannot be applied to large models (e.g. CIFAR100, TinyImageNet models) due to its high computational cost. Thus, in many settings of our paper, GCP-CROWN cannot bring improvements, yet our method still can.
>
> We evaluate the total number of branches (less is better) on several benchmarks below. GCP-CROWN+BIGs could further reduce the number of branches compared with GCP-CROWN only. Note that GCP-CROWN cannot scale to CIFAR100 and TinyImageNet, yet our bound implications can still bring improvements. We added these results to Appendix G.
>
>
> |            | $\beta$-CROWN | GCP-CROWN     | BIG     | GCP-CROWN+BIG     |
> |-------------|---------|---------|---------|-------------|
> | MNIST-A-Adv | 53120.9 | 22250.4 | 45130.4 | **21230.4** |
> | CIFAR-A-Adv | 50614.2 | 24103.7 | 46132.4 | **23792.5** |
> | CIFAR-A-Mix | 55764.8 | 28607.1 | 50014.2 | **28204.6** |
> | TinyImageNet-Medium | 64253.9 | - | 60082.5 | - |
> | CIFAR100-small | 74113.5 | - | 70006.6 | - |
>
>
> > Did you investigate the interaction of these bound implications with the general cutting planes from GCP (also based on beta-CROWN)?
>
> That’s a very interesting question. We exported the cutting planes from GCP-CROWN and compared them to our implications on the same set of instances. We report the average number of implications found by us, the average number of cutting planes in GCP-CROWN, and the overlaps. Here overlap means that GCP-CROWN finds a constraint mentioning the same pair of two neurons as we found as implicated and implicant neurons.
>
> | Average Implications | GCP-CROWN |   BIG  | Overlaps |
> |----------------------|:---------:|:------:|:------:|
> | MNIST-A-Adv          |   574.2   |  761.7 |  102.2 |
> | CIFAR-A-Adv          |   683.7   | 1103.3 |  89.9  |
> | CIFAR-A-Mix          |   796.9   | 1289.7 |  104.2 |
>
> As we can see, the overlaps only take a minority of the overall implications we found with our method. This is expected because GCP-CROWN uses a MIP solver to generate cutting planes, which is unaware of the underlying neural network architecture and uses generic methods (such as Gomory cuts) to find implications among variables. We added these results to Appendix G.
>
> > Significance of improvements on established benchmarks
>
> Thanks for mentioning the issue. We believe that minor improvements on established benchmarks are largely due to the benchmark design. These benchmarks were proposed when neural network verifiers were relatively weak. Nowadays a majority of SDP-FO[1] and ERAN[2] benchmarks have been largely solved, while a small portion of unsolved instances with unknown (potentially very high) difficulty. The improvement in the verification framework cannot be fully reflected by considering these instances with unknown difficulty. Our new VeriHard benchmark contains difficult examples that existing verifiers can guarantee to solve with a long timeout (so the difficulty is known, similar to the classic oval20 benchmark used in many papers). The progress of neural network verifiers can be reflected better in this new benchmark.
>
> [1] Dathathri, Sumanth, et al. "Enabling certification of verification-agnostic networks via memory-efficient semidefinite programming." Advances in Neural Information Processing Systems 33 (2020): 5318-5331.
>
> [2] Singh, Gagandeep, et al. "Fast and effective robustness certification." Advances in neural information processing systems 31 (2018).

---

> ### Author Response · Authors · 2023-11-22
> **We added GCP-CROWN and branching heuristic results as you suggested and improved clarity (Part 2/2)**
>
> > Presentation of Theorem 3.2 can be improved. An illustration of Theorem 3.2 could help communicate the underlying intuition.
>
> We have split Theorem 3.2 into a Theroem and a Corollary, as suggested. Here “linear bounds to not have an intersection with C” means that the implicant lower constraint Eq. (6) is too far away from $x_0$, and it is outside of the box C centered at $x_0$. We have revised the presentation of the theorem accordingly. In addition, **we’ve added a figure** (Fig. 2) to showcase the cases in the Corollary, as you suggested.
>
> > The scaling of the y-axis in Figure 2 is unclear making it hard to interpret. I would consider a linear scale instead.
>
> Thanks for pointing this out. We revised our figures to have a linear y-axis scale in our revision.
>
> > I believe it should be "(inputs <= 0)" for the "inactive" case in Section 1 paragraph 3. Other typos.
>
> Thanks for your nice catch! We have fixed these typos in our revision.

---

> > ### Comment · Reviewer_2ULS · 2023-11-22
> > **Thank you for the response!**
> >
> > I want to thank the authors for addressing most of my questions and taking into account my suggestions on how to improve the clarity of presentation. However, I have two remaining concerns:
> >
> > **Improvement over GCP**
> > Indeed, while the overlap with GCP constraints seems small (using the authors' metric), the reduction in considered branches is heavily reduced. Further, the two settings to which GCP does not scale, show much less of an improvement both in terms of considered subproblems and verified accuracy even on the VeriHard benchmark. Finally, I believe reporting the certified accuracy of GCP as a baseline in Table 2 remains crucial.
> >
> > **Inadequate comparison to related work**
> > More importantly, I believe the novelty concerns raised by Reviewer HQLe can and should not be addressed within a rebuttal. The authors title their response with "We revised our entire paper", which, I believe, suggests it should undergo a new review cycle. I have thus updated my score accordingly.

---

> > > ### Author Response · Authors · 2023-11-22
> > > **We thank you for your quick response. We want to point out a few major misunderstandings**
> > >
> > > We thank you for your quick response. We want to point out a few **major misunderstandings**. We hope you can reconsider your evaluation of our paper.
> > >
> > > > The authors title their response with "We revised our entire paper", which, I believe, suggests it should undergo a new review cycle.
> > >
> > > When we said we revised the “entire” paper, it meant that we made some revisions in all sections. **We have clearly marked these revisions in blue**. We are not rewriting the paper. There are a few sentences added in each section (clearly marked). In fact, **many ICLR submissions make similar revisions to their papers** (and it is a great advantage of the OpenReview system). Saying that making these changes would require “a new review cycle” is unfair to us.
> > >
> > >
> > > > Inadequate comparison to related work
> > >
> > > We have added additional comparisons to the papers requested by Reviewer HQLe. Our method **outperformed the baseline by 7x - 170x more** in new results in Table 1, thanks to the stronger theoretical formulation. In fact, **even in the original version of our paper, we already had comparisons to the tool Venus2** (in Table 3), but we unfortunately did not correctly cite and discuss all the papers behind the tool, which caused the complaint. We humbly provided a lot of new comparisons, both theoretically and experimentally, to reviewer HQLe, mostly because we want to make the reviewer aware that we know the importance of these papers, and we have correctly cited and discussed these papers now so hopefully reviewer HQLe can feel more comfortable. **Our humbleness should not be interpreted as a weakness of our work**.
> > >
> > > > Improvement over GCP
> > >
> > > We want to point out that we have made our best effort to implement our method in GCP-CROWN within a very short period. Due to the limited time, our code efficiency is not comparable to the original optimal implementation, so the improvement is limited. However, we have shown that the bound implications we found are largely non-overlapping, and there is a potential for good improvement. We must point out that our main contribution is a novel method to find bound implications and it is a generic strategy, and **we cannot feasibly implement it into every verifier**. The bound implications we found are already **one to two orders of magnitude more compared to previous papers** pointed out by reviewer HQLe. Even in these previous papers, they only implemented their method in one baseline and not all the baselines.
> > >
> > > We appreciate all the constructive feedback from you again, and we hope you can reasonably reevaluate our paper. Thank you.

---

> > > > ### Comment · Reviewer_2ULS · 2023-11-23
> > > > **Follow Up**
> > > >
> > > > I thank the authors for their candid response. I have again reviewed the author's response to Reviewer HQLe and all (highlighted) changes to the paper. While I believe that the changes made go beyond those typical for ICLR and crucially affect the novelty and contribution of this paper, the proposed method still seems promising and empirically and theoretically sufficiently distinct from prior work. I am thus willing to raise my score again and leave the judgment of whether changes of this magnitude require another review cycle to the AC.
> > > >
> > > > **Comparison to GCP**
> > > > While the authors can certainly not be asked to implement their approach into every verifier, this is not what I am asking them to do. I believe the proposed method can be presented in two ways:
> > > > * As a way to construct **bound implication graphs**, irrespective of the used verifier. In this case, I would expect the bound implication graphs discovered by prior work to be integrated into the same classifier and evaluated on equal footing (going beyond a pure numbers comparison). To make a strong case for the relevance of the contribution I would appreciate a demonstration that the result verifier beats the current state-of-the-art.
> > > > * As a new **verifier** in which case I believe comparing against the state-of-the-art (including GCP) is essential. In this case an integration in the state-of-the-art verifier would certainly help with a positive performance comparison *if* the methods are truly complimentary.
> > > >
> > > > However, I believe the authors have not substantiated either case. They do not compare against the strongest available verifier (GCP, which is a different version of $\alpha\beta$-CROWN that they build on), nor do they include a fair comparison to other bound implication graphs.

---

### Official Review · Reviewer_2T3h · 2023-11-01

**Soundness:** 4 excellent
**Presentation:** 4 excellent
**Contribution:** 4 excellent
**Rating:** 8
**Confidence:** 3

**Summary:**

The paper proposes a novel method named bound implication graph (BIG) to efficiently reduce the number of subproblems when verifying neural networks with the bound-and-branch techniques. The bound implication graph is constructed to capture the neuron dependencies with pre-computed estimated bound on each neuron and track the active/inactive status of the neuron given the branch status of the other neuron. The experiments demonstrate a significant reduction in verification time and an improvement in the tightness of bounds.

**Strengths:**

- The paper is sound and the topic of the paper is of high interest to the research community.
- The paper does a good job at introducing technical details and makes the paper easy to follow.
- As far as I know, the paper is the first to provide ways to efficiently reduce the branches for verifying neural networks by explicitly tracking the neuron dependencies as active/inactive states.
- The empirical study shows great improvement of the proposed method over existing verification algorithms in both verification time and verification results.
- Comprehensive ablation studies on the studied problems.
- The paper also provides a new dataset as VeriHard which contains instances that can only be solved using state-of-the-art verifiers with a long timeout.

**Weaknesses:**

- More discussions on the bottleneck of the proposed method would further strengthen the paper.

**Questions:**

- Could you comment on how would the BIG can be used to discover hard adversarial examples or train the model to be robust against those examples?

---

> ### Author Response · Authors · 2023-11-22
> **We greatly appreciate your positive comments, and we answer your questions below.**
>
> Thanks for recognizing the importance of our work and for strongly supporting our paper! Following your suggestions, we’ve added discussions of bottlenecks to the paper and also considered using BIG to discover hard adversarial examples.
>
> > More discussions on the bottleneck of the proposed method would further strengthen the paper.
>
> Thanks for your suggestion. We’ve added a discussion to our paper (Section H) on the bottleneck as suggested. Our proposed method is highly relied on BaB-based verification framework so it has some common issues: 1) Our derived implications follow the large body of papers for verifying ReLU networks [1,2,3,4] and need further extensions to other non-piecewise linear activation functions. 2) We cannot support the verification of very large neuron networks, such as Transformer with billions of parameters. We will keep investigating the possibility of applying our implication concept into the verification of more general models in the future.
>
> > Could you comment on how would the BIG can be used to discover hard adversarial examples or train the model to be robust against those examples?
>
> This is a very interesting direction. In our work, we did not use our implication graph (BIG) to discover adversarial examples during branch-and-bound, but there exist some papers focusing on finding adversarial examples through branch-and-bound procedure [5]. Our method can be applied on top of these BaB-based methods to reduce the research space when finding adversarial examples, and this could be an interesting future work.
>
> [1] Katz, Guy, et al. "The marabou framework for verification and analysis of deep neural networks." Computer Aided Verification: 31st International Conference, CAV 2019, New York City, NY, USA, July 15-18, 2019, Proceedings, Part I 31. Springer International Publishing, 2019.
>
> [2] Wang, Shiqi, et al. "Beta-crown: Efficient bound propagation with per-neuron split constraints for neural network robustness verification." Advances in Neural Information Processing Systems 34 (2021): 29909-29921.
>
> [3] Ferrari, Claudio, et al. "Complete verification via multi-neuron relaxation guided branch-and-bound." arXiv preprint arXiv:2205.00263 (2022).
>
> [4] Zhang, Huan, et al. "General cutting planes for bound-propagation-based neural network verification." Advances in Neural Information Processing Systems 35 (2022): 1656-1670.
>
> [5] Zhang, Huan, et al. "A branch and bound framework for stronger adversarial attacks of relu networks." International Conference on Machine Learning. PMLR, 2022.

---

### Meta-Review · Area_Chair_AKjF · 2023-12-05

**Metareview:**

This work presents an improved method for branch-and-bound verification of ReLU neural networks. The key insight rests in building an implication graph based on the consequences of a certain collection of neurons firing (or not firing). I find that the work is interesting and likely to be a useful direction for the community.

That being said, the reviewers were split on whether to accept the work. By majority, reviewers (including several domain experts) rate the work as an incremental improvement over prior work, conceptually, which should have required the Authors to more carefully compare and contextualise their contributions with further experiments, especially with respect to their method's complementarity to GCP.

Reviewer 2ULS, in their latest response to the Authors, identified two concrete ways in which the paper's presentation could have been improved, which both sound sensible to me. While the Authors did respond by including a section on GCP in the Appendix, it is evident from the discussion and the organisation of the paper that more work could have been done to properly elaborate on the complementarity with GCP. There is not a clear sign of strong improvements w.r.t. GCP on the tasks where GCP does terminate on time, the Authors stated that they did not have the time to properly implement GCP themselves, which casts, in my opinion, reasonable doubt about the robustness of these results. Coupled with the fact that all results on GCP do not feature in the main paper -- and it seems clear that GCP _should_ be discussed more prominently, regardless of the route the Authors take -- my ultimate conclusion is that the paper would absolutely benefit from another, careful, round of revisions, giving the Authors more time to properly incorporate and strengthen their message.

The Reviewers, by majority -- including one of the accepting reviewers -- support this assessment and encourage the authors to revise and resubmit their work to a future venue. Further, none of the reviewers opted to champion the work or oppose the decision in its present form.

While the work presents a contribution which has clear potential, it is evident to me that the paper would strongly benefit from another revision cycle, and publishing it in its present form would diminish its significance. I wish the Authors every success in their future submission of the work!

_[I stress that, while the Authors did indeed revise their paper significantly during the Rebuttal, I found that Reviewer 2ULS' remark about the magnitude of changes to the work to be somewhat excessive, and have not taken this remark into account when writing in this meta-review.]_

**Justification For Why Not Higher Score:**

While the work presents a contribution which has clear potential, it is evident to me that the paper would strongly benefit from another revision cycle, and publishing it in its present form would diminish its significance. The Reviewers, by majority, support this conclusion, and none have opposed it.

**Justification For Why Not Lower Score:**

N/A

---

### Decision · Program_Chairs · 2024-01-16

Reject